# Entanglement entropies of minimal models from null-vectors

### Thomas Dupic⋆, Benoit Estienne and Yacine Ikhlef

Sorbonne Université, CNRS, Laboratoire de Physique Théorique et Hautes Energies, LPTHE,
F-75005 Paris, France

⋆ tdupic@lpthe.jussieu.fr

## Abstract

We present a new method to compute Rényi entropies in one-dimensional critical systems. The null-vector conditions on the twist fields in the cyclic orbifold allow us to derive a differential equation for their correlation functions. The latter are then determined by standard bootstrap techniques. We apply this method to the calculation of various Rényi entropies.



# 1  Introduction

Ideas coming from quantum information theory have provided invaluable insights and powerful tools for quantum many-body systems. One of the most basic tools in the arsenal of quantum information theory is (entanglement) entropy [1]. Upon partitioning a system into two subsystems, $\mathscr{A}$ and $\mathscr{B}$, the entanglement entropy is defined as the von Neumann entropy $S(\mathscr{A}) = -\text{Tr}\,\rho_{\mathscr{A}} \log \rho_{\mathscr{A}}$, with $\rho_{\mathscr{A}}$ being the reduced density matrix of subsystem $\mathscr{A}$.

Entanglement entropy (EE) is a versatile tool. For a gapped system in any dimension, the entanglement entropy behaves similarly to the black hole entropy : its leading term grows like the area of the boundary between two subsystems instead of their volume, in a behavior known as the *area law* [2–8] :

$$S(\mathscr{A}) \simeq \alpha \, \text{Vol}(\partial \mathscr{A}) \,,$$

where $\alpha$ is a non-universal quantity. Quantum entanglement – and in particular the area law – has led in recent years to a major breakthrough in our understanding of quantum systems, and to the development of remarkably efficient analytical and numerical tools. These methods, dubbed *tensor network* methods, have just begun to be applied to strongly correlated systems with unprecedented success [9–11].

For critical systems, a striking result is the universal scaling of the EE in one-dimension [12–14]. For an infinite system, with the subsystem $\mathscr{A}$ being a single interval of length $\ell$, one has

$$S([0, \ell]) \simeq \frac{c}{3} \log \ell \,,$$

where $c$ is the central charge of the underlying Conformal Field Theory (CFT). This result is based on a CFT approach to entanglement entropy combined with the replica trick, which maps the (Rényi) EE to the partition function on an $N$-sheeted Riemann surface with conical singularities. In some particular cases – essentially for free theories – it is possible to directly calculate this partition function [15–20] using the general results from the 1980's for free bosonic partition functions on Riemann surfaces [21–24]. In most cases however this is very difficult. An alternative approach is to replicate the CFT rather than the underlying Riemann surface. Within this scheme one ends up with the tensor product of $N$ copies of the original CFT modded out by cyclic permutations, and the conical singularities are mapped to twist fields, denoted as $\tau$. These theories are known as *cyclic orbifolds* [25–28]. Within this framework the Rényi EE boils down to a correlation function of twist fields in the cyclic orbifold [29,30]. The case of a single interval is particularly simple as it maps to a two-twist correlation function. When the subsystem $\mathscr{A}$ is the union of $m > 1$ disjoint intervals most results are restricted to free theories [16–20,31], and much less is known in general [32]. In the orbifold framework, this maps to a $2m$-twist correlation function, which is of course much more involved to compute than a simple two-point function.

In this article we report on a new method to compute twist fields correlation functions. Our key ingredients are (i) the null-vector conditions obeyed by the twist fields under the extended algebra of the cyclic orbifold and (ii) the Ward identities obeyed by the currents in this extended algebra. Note that the null-vector conditions for twist fields were already detected in [26], but until now then they have only been exploited to determine their conformal dimension. Our method is quite generic, the only requirement being that the underlying CFT be rational (which in turn ensures that the induced cyclic orbifold is rational). This approach provides a rather versatile and powerful tool to compute the EE that is applicable to a variety of situations, such as non-unitary CFT, EE of multiple intervals, EE at finite temperature and finite size, and/or EE in an excited state.

We illustrate this method with the most basic minimal model of CFT: the Yang-Lee model. This model has only two primary fields: the identity $\mathbb{1}$ and the field $\phi$. However, the simplicity of this situation – in particular, the nice form of the null-vector conditions obeyed by the identity operator – comes with a slight complication: the model is not unitary, and $\phi$ has a negative dimension. Hence, the vacuum $|0\rangle$ and the ground state $|\phi\rangle$ are distinct (i.e. the vacuum is not the state with lowest energy), which implies that the ground state breaks conformal invariance. This leads to an important modification in the path integral description used in the replica trick : the boundary conditions at every puncture must reflect the insertion of the field $\phi$ (and not the identity operator). In practice, this means that the twist field $\tau$ must be replaced by $\tau_\phi \propto\, :\tau\phi:$ as noted in [33], but also that the correlation functions of these twist fields must be evaluated in the ground state $|\phi\rangle$ rather than in the vacuum $|0\rangle$.

Hence we see that, for the Yang-Lee model at finite size, even the single-interval entropy requires the computation of a four-point function, and this is where the full power of null-vector equations can be brought to bear.

The plan of this article is as follows. In Sec. 2 we review the cyclic orbifold construction of [26–28], and its relation to Rényi entropies. In Sec. 4 we describe a basic example where the null-vector conditions on the twist field only involve the usual Virasoro modes, and thus yield straightforwardly a differential equation for the twist correlation function. In Sec. 5, we turn to more generic situations, where the null-vector conditions involve fractional modes of the orbifold Virasoro algebra: we first introduce the Ward identities for the conserved currents

$\widehat{T}^{(r)}(z)$ in the cyclic orbifold, and use them to derive differential equations for a number of new twist correlation functions. Finally, in Sec. 6 we describe a lattice implementation of the twist fields in the lattice discretisation of the minimal models, namely the critical Restricted Solid-On-Solid (RSOS) models. We conclude with a numerical check of our analytical results for various EEs in the Yang-Lee model.

## 2 General background

### 2.1 Entanglement entropy and conformal mappings

Consider a critical one-dimensional quantum system (a spin chain for example), described by a conformal field theory (CFT). Suppose that the system is separated into two parts : $\mathscr{A}$ and its complement $\mathscr{B}$. The *amount of entanglement* between $\mathscr{A}$ and $\mathscr{B}$ is usually measured through the Von Neumann entropy. If the system is in a normalised pure state $|\psi\rangle$, with density matrix $\rho = |\psi\rangle\langle\psi|$, its Von Neumann entropy is defined as:

$$S(\mathscr{A}, \psi) = -\mathrm{Tr}_{\mathscr{A}}[\rho_{\mathscr{A}} \log(\rho_{\mathscr{A}})], \qquad \text{where} \quad \rho_{\mathscr{A}} = \mathrm{Tr}_{\mathscr{B}}|\psi\rangle\langle\psi|. \tag{1}$$

The Rényi entropy is a slight generalisation, which depends on a real parameter $N$:

$$S_N(\mathscr{A}, \psi) = \frac{1}{1-N} \log \mathrm{Tr}_{\mathscr{A}}(\rho_{\mathscr{A}}^N). \tag{2}$$

In the limit $N \to 1$, one recovers the von Neumann entropy: $S_{N\to 1}(\mathscr{A}, \psi) = S(\mathscr{A}, \psi)$.

For integer $N$, a replica method to compute this entropy was developed in [14] (see [29] for a recent review). The main idea consists in re-expressing geometrically the problem. The partial trace $\rho_{\mathscr{A}}$ acts on states living in $\mathscr{A}$ and propagates them, while tracing over the states in $\mathscr{B}$. It can be seen as the density matrix of a "sewn" system kept open along $\mathscr{A}$ but closed on itself elsewhere.

When $\mathscr{A}$ is a single interval ($\mathscr{A} = [u, v]$), the resulting Riemann surface is conformally equivalent to the sphere. It can be *unfolded* (mapped to the sphere) using a change of variable of the form:

$$w = \left(\frac{z-u}{z-v}\right)^{1/N}. \tag{3}$$

When $|\psi\rangle = |\mathrm{vac}\rangle$ is the vacuum state of the CFT, this change of coordinates allows [29] to compute the entropy of a single interval in an infinite system, with the well-known result:

$$S_N([u, v], \mathrm{vac}) = \frac{c}{6} \frac{N+1}{N} \log |u-v|. \tag{4}$$

Throughout this paper, we shall rather consider the case of an interval $\mathscr{A} = [0, \ell]$ in a finite system of length $L$ with periodic boundary conditions. In this case, one has [29]:

$$S_N(\ell/L, \mathrm{vac}) = \frac{c}{6} \frac{N+1}{N} \log\left[ L \sin\left(\frac{\pi \ell}{L}\right) \right], \tag{5}$$

where we have slightly changed the notation to indicate that the total system is of finite size $L$.

This type of calculations becomes more complicated for the entropy of other states than the vacuum: two operators then need to be added on each of the sheets of $\Sigma_N$. This is one of

the main limitations of the method based on conformal mapping : a lot of the structure of the initial problem disappears after the conformal map. In this case, a one-variable problem (the size of the interval) becomes a $2N$-variable problem. These complicated correlation functions have only been computed for free theories [34, 35], and have been used in various contexts since then [36–38]. Moreover, if $\Sigma$ is the initial surface where the system lives, then the genus of $\Sigma_N$ is $g(\Sigma_N) = Ng(\Sigma) + (N-1)(p-1)$, where $p$ is the number of connected components of $\mathscr{A}$. Hence if $\mathscr{A}$ is not connected or if the initial surface is not the Riemann sphere, one has to deal with CFT on higher-genus surfaces.

## 2.2 Correlation functions of twisted operators

The Rényi entropies can alternatively be interpreted as correlation functions of twist operators. We consider a system of finite length $L$ with periodic boundary conditions, in the quantum state $|\psi\rangle$. In the scaling limit, this corresponds to a CFT on the infinite cylinder of circumference $L$, with boundary conditions specified by the state $\psi$ on both ends of the cylinder. After the conformal mapping $z \mapsto \exp\frac{2\pi z}{L}$, one recovers the plane geometry. The Rényi entropy of a single interval $\mathscr{A} = [0, \ell]$ in the pure state $|\psi\rangle$ is given as a correlation function in the $\mathbb{Z}_N$ orbifold CFT (see below):

$$S_N([0,\ell], \psi) = \frac{1}{1-N} \log\langle\Psi|\tau(1)\widetilde{\tau}(x,\bar{x})|\Psi\rangle, \qquad x = \exp(2i\pi\ell/L), \qquad (6)$$

where $\Psi = \psi^{\otimes N}$ corresponds to $N$ replicas of the operator $\psi$ at a given point. The twist operators $\tau$ and $\widetilde{\tau}$ implement the branch points a the ends of the interval. Since these branch points introduce singularities in the metric, one has to choose a particular regularisation of the theory at each branch point: each choice of regularisation corresponds to a choice of primary twist operator $\tau$. The classification of primary twist operators is obtained by the induction procedure (see Sec. 3.3), which uniquely associated any primary operator $\phi$ of the mother CFT to a twist operator $\tau_\phi$, with dimension $\widehat{h}_\phi = (N - 1/N)c/24 + h_\phi/N$. In a unitary CFT, the most relevant operator is the identity (i.e. the conformally invariant operator), and the correct choice for the twist operator in (6) is $\tau = \tau_{\mathbb{1}}$. In Sec. 6 we introduce the construction of a lattice regularisation scaling to *any* given primary twist operator $\tau_\phi$ in a minimal model of CFT.

More generally, if $\mathscr{A}$ is a union of $p \geq 1$ disjoint intervals:

$$\mathscr{A} = [u_1, v_1] \cup [u_2, v_2] \cup \ldots [u_p, v_p],$$

then one may define the *p*-interval correlation function:

$$\langle\Psi|\tau_1(y_1, \bar{y}_1)\widetilde{\tau}_1(x_1, \bar{x}_1)\ldots\tau_p(y_p, \bar{y}_p)\widetilde{\tau}_p(x_p, \bar{x}_p)|\Psi\rangle, \qquad (7)$$

with

$$x_j = \exp(2i\pi v_j/L), \qquad y_j = \exp(2i\pi u_j/L),$$

and any choice of twist operators $(\tau_1, \ldots \tau_p)$ and $(\widetilde{\tau}_1, \ldots \widetilde{\tau}_p)$.

## 2.3 Non-unitary models

Although the goal of the present paper is not to study specifically entanglement in non-unitary models, some emphasis is put on the Yang-Lee singularity model. The reason for this is that the corresponding minimal model has the simplest operator algebra (it has only two primary fields), which makes calculations more tractable and easy to present. However it should be stressed that what we are computing are partition functions on $N$-sheeted surfaces. For a unitary model this corresponds to Rényi entropies, and for that reason we chose to refer to these

partition functions as "entropies" even in the non-unitary case. This is just a matter of terminology, and we do not claim that they provide a good measure of the amount of entanglement.

The problem of entanglement entropy in non-unitary models has already been addressed in various contexts [33, 39–41]. For comparison with the existing literature on the subject, we clarify in this section the specific choices and observations that we made for non-unitary models. We refrain from using the bra/ket notations to avoid any possible source of confusion.

Consider a Hamiltonian $H$ acting on a vector space $E$. The transpose operator ${}^t H$ acts in the dual space (consisting of all linear forms) $E^*$ as

$$
{}^t H(w) = w \circ H , \tag{8}
$$

for any linear form $w$. We assume that $H$ is diagonalizable with a discrete spectrum and eigenbasis $\{r_j\}$

$$
H r_j = E_j r_j . \tag{9}
$$

The dual basis $\{w_j\}$, which is defined by $w_i(r_j) = \delta_{ij}$, is an eigenbasis of ${}^t H$

$$
w_j \circ H = E_j w_j , \tag{10}
$$

and the Hamiltonian can be written as

$$
H = \sum_j E_j r_j w_j . \tag{11}
$$

A possible definition for the density matrix of the system at inverse temperature $\beta$ is

$$
\rho = \frac{1}{Z} e^{-\beta H} = \frac{1}{Z} \sum_j e^{-\beta E_j} r_j w_j, \qquad Z = \sum_j e^{-\beta E_j} . \tag{12}
$$

In particular at zero temperature this yields

$$
\rho = r_0 w_0 , \tag{13}
$$

where $r_0$ denotes the ground state of $H$. Assuming a decomposition $E = E_{\mathscr{A}} \otimes E_{\mathscr{B}}$ one can then trace over $\mathscr{B}$ to define $\rho_{\mathscr{A}}$. Let $\{f_j\}$ be a basis of $E_{\mathscr{B}}$ and $\{f_j^*\}$ the dual basis, the trace over $\mathscr{B}$ is defined as

$$
\rho_{\mathscr{A}} = \text{Tr}_{\mathscr{B}}(\rho) = \sum_j \left( \mathbb{1}_{\mathscr{A}} \otimes f_j^* \right) \rho \left( \mathbb{1}_{\mathscr{A}} \otimes f_j \right) . \tag{14}
$$

Note that tracing over $\mathscr{B}$ is independent of the basis $\{f_j\}$ chosen, and does not require any inner product. With $\rho_{\mathscr{A}}$ at hand one then defines the Von Neumann and Rényi entropies in the usual way.

The main advantage of this construction is that the corresponding (Rényi) entropy $\text{Tr}(\rho_{\mathscr{A}}^N)$ maps within the path-integral approach to an Euclidean partition function on an $N$-sheeted Riemann surface. Underlying this result is the identification of the reduced matrix $\rho_{\mathscr{A}}$ with the partition function on a surface leaving open a slit along $\mathscr{A}$. Note that such a partition function can be computed purely in terms of matrix elements of the transfer matrix, and therefore it does not involve any inner product structure.

The disadvantages of this construction are twofold. The main one is that the reduced density matrix (and hence the entanglement entropy) may not be positive. While this may

seem like a pathological property, loss of positivity in a non-unitary system might be acceptable depending on the context and motivations. The other one is that this definition only applies to eigenstates of $H$ (and statistical superposition thereof). This stems for the fact that there is no canonical (i.e. basis independent) isometry between $E$ and $E^*$. In practice this means that knowing the ground state $r_0$ is not enough to compute the entanglement entropy, one also needs to know $H$ to characterize $w_0$.

Consider now an inner-product structure on $E$, *i.e.* a non-degenerate hermitian form[1] $\langle \cdot, \cdot \rangle$. By virtue of being non-degenerate, this inner product induces a canonical isometry between linear forms and vectors. For every vector $v \in E$, denote by $v^\dagger$ the linear form defined by

$$v^\dagger(x) = \langle v, x \rangle, \qquad x \in E.$$

Every element in $E^*$ can be written in this form, and the map $I : v \to v^\dagger$ is an antilinear isometry from $E$ to $E^*$. In particular one can associate a vector $l_j$ to every linear form $w_j$ such that $w_j = l_j^\dagger$. The vectors $l_j$ are what is commonly referred to as left eigenvectors of $H$. These are nothing but the eigenvectors of $H^\dagger$, the hermitian adjoint of $H$, which is characterized by the following property

$$\langle H^\dagger v_1, v_2 \rangle = \langle v_1, H v_2 \rangle$$

for any vectors $v_1, v_2$ in $E$. The transpose ${}^t H$ and the Hermitian adjoint $H^\dagger$ are closely related:

$$H^\dagger = I^{-1} \circ {}^t H \circ I.$$

In particular the relation ${}^t H(w_j) = E_j w_j$ becomes

$$H^\dagger l_j = E_j^* l_j. \tag{15}$$

While the previous prescription for the density matrix amounts in this context to

$$\rho = r_0 l_0^\dagger, \tag{16}$$

an alternative prescription is

$$\tilde{\rho} = r_0 r_0^\dagger. \tag{17}$$

For many non-unitary models there exist a natural notion of inner product that makes the Hamiltonian self-adjoint and is compatible with locality,[2] at the cost of not being definite-positive. For such an inner-product left and right eigenvectors coincide and it follows that

$$\rho = \tilde{\rho}.$$

This is typically the case within the CFT framework : the standard CFT inner product is such that $L_n^\dagger = L_{-n}$, and in particular the Hamiltonian is self-adjoint: $L_0 = L_0^\dagger$. This is also the case for the loop model based on the Temperley-Lieb algebra [41] or for the Yang-Lee spin chain (see appendix E).

Let us now assume that the inner product is definite-positive. For a unitary system $H = H^\dagger$: left and right eigenvectors coincide, and both prescriptions yield the usual notion of density

---

[1]An inner product is usually required to be positive definite. At this point we do not make this assumption, but we will come back to this when discussing the positivity of the reduced density matrix.

[2]In the sense that the Hamiltonian density is self-adjoint.

matrix and entanglement. For a non-Hermitian Hamiltonian operator $H$ in general $l_j$ and $r_j$ are different (even if $E_j$ is real). If $H$ is symmetric but not real (in some orthonormal basis), then the eigenvectors have non-real components, and are related through complex conjugation:

$$l_j = r_j^*.$$

On a more fundamental level this illustrates the fact that the canonical map between linear forms and vectors is antilinear.

The prescription $\tilde{\rho}$, together with a positive-definite inner product, seems to be physically more natural than $\rho$ as it yields a positive entanglement entropy (as can be seen from the Schmidt decomposition). Moreover it does not depend on $H$, only on the state considered and on the inner product. However this quantity is very much sensitive to the inner product chosen, and for a non-Hermitian Hamiltonian there is no canonical choice of a positive-definite inner product.

On a more technical side, when computing any quantity involving $\tilde{\rho}$ in the path-integral formalism one needs to implement explicitly the (inner-product dependent) time-reversal operation $r_0 \to l_0$ (e.g. $r_0 \to r_0^*$ for symmetric $H$) in order to get a consistent Euclidean description. Such a time-reversal defect can be thought of as a specific boundary condition in the tensor product $CFT^{\otimes 2}$, which is typically a difficult problem.

It has been argued in [33] that for $PT$-symmetric Hamiltonian, the left and right ground states $r_0$ and $l_0$ coincide (while working with a definite positive inner product). This would circumvent this difficulty. However we found that this is not the case for the Yang-Lee model in finite size. Moreover assuming $r_0 = l_0$ immediately yields positive entanglement entropies, which again we found is not the case (both within our numerical and analytical calculations, see Figure 3).

In the following, when the model considered is non-unitary, we will choose (13) as the density matrix so that the Euclidean path-integral formalism described in Sec. 2.1, i.e. the interpretation of Rényi entropies as partition functions on a replicated surface with branch points, can be used straightforwardly. This is also the choice made in [39,41].

Within the Euclidean path-integral formalism an additional fact to take into account when studying non-unitary models is the existence of a primary state $\phi$ with a conformal dimension lower than the CFT vacuum $h = 0$ (i.e. the conformally invariant state). As was first pointed out in [33], this has a dramatic effect on the twist field : the most relevant twist operator is no longer $\tau_{\mathbb{1}}$, but rather $\tau_\phi$. Repeating the steps of section 2.2, the one-interval Rényi entropy in the ground state $|\phi\rangle$ is mapped within the orbifold approach to

$$\mathrm{Tr}(\rho_{\mathscr{A}}^N) = \langle\Phi|\tau_\phi(u)\widetilde{\tau}_\phi(v)|\Phi\rangle, \tag{18}$$

where $\Phi = \phi^{\otimes N}$. In [33] it was further claimed that the entanglement entropy in a non-unitary model behaves as

$$S_N \sim \frac{c_{\mathrm{eff}}}{6}\frac{N+1}{N}\log|u-v|, \tag{19}$$

where $c_{\mathrm{eff}} = c - 24h_\phi$ is the effective central charge. However this result was based on an incorrect mapping to an Euclidean partition function, namely

$$\mathrm{Tr}(\rho_{\mathscr{A}}^N) = \frac{\langle\tau_\phi(u)\widetilde{\tau}_\phi(v)\rangle}{\langle\Phi(u)\Phi(v)\rangle} \tag{20}$$

instead of (18). We claim that the behavior (19) is incorrect, and the Cardy-Calabrese formulas (4–5) for the entanglement entropy[3] cannot be applied, even with the substitution $c \rightarrow c_{\text{eff}}$.

# 3  The cyclic orbifold

The expression (4) suggests that the partition function on $\Sigma_N$ can be considered as the two-point function of a "twist operator" of dimension

$$h_\tau = \frac{c}{24}\left(N - \frac{1}{N}\right).$$
(21)

Indeed, this point of view corresponds to the construction of the cyclic orbifold. Mathematically, one starts with $N$ copies of the original CFT model (called the *mother CFT*, with central charge $c$, living on the original surface $\Sigma$) then mod out the $\mathbb{Z}_N$ symmetry[4] (the cyclic permutations of the copies). This cyclic orbifold theory was studied extensively in [26–28]. We give an overview of the relevant concepts that we shall use.

## 3.1  The orbifold Virasoro algebra

All the copies of the mother CFT have their own energy-momentum tensor $T_j(z)$ [and $\bar{T}_j(\bar{z})$ for the anti-holomorphic part]. Their discrete Fourier transforms (in replica space) are called $\widehat{T}^{(r)}(z)$, $r \in \{0, \cdots, N-1\}$ and are defined by:

$$\widehat{T}^{(r)}(z) = \sum_{j=1}^{N} e^{2i\pi r j/N} T_j(z), \qquad T_j(z) = \frac{1}{N} \sum_{r=0}^{N-1} e^{-2i\pi r j/N} \widehat{T}^{(r)}(z).$$
(22)

The currents $T_j$ are all energy-momentum tensors of a conformal field theory, so their Operator Product Expansion (OPE) with themselves is:

$$T_j(z) T_k(0) = \delta_{j,k}\left[\frac{c/2}{z^4} + \frac{2T_j(z)}{z^2} + \frac{\partial T_j(z)}{z}\right] + \text{regular terms}.$$
(23)

For two distinct copies, $T_j(z_1) T_k(z_2)$ is regular ; on the unfolded surface, even when $z_1 \rightarrow z_2$ the two currents are at different points. With that in mind, the OPE between the Fourier transforms of these currents can be written:

$$\widehat{T}^{(r)}(z) \widehat{T}^{(s)}(0) = \frac{(Nc/2)\delta_{r+s,0}}{z^4} + \frac{2\widehat{T}^{(r+s)}(z)}{z^2} + \frac{\partial \widehat{T}^{(r+s)}(z)}{z} + \text{regular terms},$$
(24)

where the indices $r$ and $s$ are considered modulo $N$. The modes of the currents are defined as:

$$\widehat{L}_m^{(r)} = \frac{1}{2i\pi} \oint dz\, z^{m+1}\, \widehat{T}^{(r)}(z).$$
(25)

In the untwisted sector of the theory the mode indices $m$ have to be integers since the operators $\widehat{T}^{(r)}(z)$ are single valued when winding around the origin. In the twisted sector however the operators $\widehat{T}^{(r)}(z)$ are no longer single-valued, and the mode indices $m$ can be fractional. Generically in the cyclic $\mathbb{Z}_N$ orbifold we have

$$m \in \mathbb{Z}/N.$$
(26)

---

[3]When discussing the result of [33] the distinction between $\rho = r_0 l_0^\dagger$ and $\rho = r_0 r_0^\dagger$ is irrelevant since they argue that $l_0 = r_0$ at criticality.

[4]For simplicity, in the following, we consider only the case when $N$ is a prime integer.

The actual values of $m$ appearing in the mode decomposition are detailed below: see (35) and (38). From the OPE (24) one obtains the commutation relations:

$$\left[\widehat{L}_m^{(r)}, \widehat{L}_n^{(s)}\right] = (m-n)\widehat{L}_{m+n}^{(r+s)} + \frac{Nc}{12} m(m^2-1)\,\delta_{m+n,0}\,\delta_{r+s,0}\,, \tag{27}$$

where $(m,n) \in (\mathbb{Z}/N)^2$. The actual energy-momentum tensor in the orbifold theory is $T_{\text{orb}}(z) = \widehat{T}^{(0)}(z)$. It generates transformations affecting all the sheets in the same way, so in the orbifold it has the usual interpretation (derivative of the action with respect to the metric). Correspondingly, the integer modes $\widehat{L}_{m\in\mathbb{Z}}^{(0)}$ form a Virasoro subalgebra. The $\widehat{T}^{(r)}(z)$ for $r \neq 0$ also have conformal dimension 2, and play the role of additional currents of an extended CFT with internal $\mathbb{Z}_N$ symmetry.

## 3.2 Operator content of the $\mathbb{Z}_N$ orbifold

**The untwisted sector** Let $z$ be a regular point of the surface $\Sigma_N$. A generic primary operator at such a regular point, which we shall call an *untwisted* primary operator, is simply given by the tensor product of $N$ primary operators $\phi_1, \ldots, \phi_N$ of dimensions $h_1, \ldots, h_N$ in the mother CFT, each sitting on a different copy of the model:

$$\Phi(z) = \phi_1(z) \otimes \phi_2(z) \otimes \ldots \phi_N(z)\,. \tag{28}$$

In the case when $\Phi$ includes at least one pair of distinct operators $\phi_i \neq \phi_j$, it is also convenient to define the discrete Fourier modes

$$\widehat{\Phi}^{(r)}(z) = \frac{1}{\sqrt{N}} \sum_{j=0}^{N-1} e^{2i\pi rj/N} \phi_{1-j}(z) \otimes \phi_{2-j}(z) \otimes \cdots \otimes \phi_{N-j}(z)\,, \tag{29}$$

where the indices are understood modulo $N$. The normalisation of $\widehat{\Phi}^{(r)}(z)$ is chosen to ensure a correct normalisation of the two-point function:

$$\langle \widehat{\Phi}^{(r)}(z_1)\widehat{\Phi}^{(-r)}(z_2)\rangle = (z_1 - z_2)^{-2h_\Phi}\,, \tag{30}$$

where $h_\Phi = \sum_{j=1}^{N} h_j$. In particular, for a primary operator $\phi$ in the mother CFT, if one sets $\phi_1 = \phi_h$ and $\phi_2 = \cdots = \phi_N = \mathbb{1}$, one obtains the *principal primary fields* of dimension $h$:

$$\widehat{\phi}_h^{(r)}(z) = \frac{1}{\sqrt{N}} \sum_{j=1}^{N} e^{2i\pi rj/N} \mathbb{1}(z) \otimes \cdots \otimes \underset{(j\text{-th})}{\phi_h(z)} \otimes \cdots \otimes \mathbb{1}(z)\,. \tag{31}$$

The OPE of the currents with generic primary operators are:

$$\widehat{T}^{(r)}(z)\widehat{\Phi}^{(s)}(0) = \frac{\widehat{h}_\Phi^{(r)}\widehat{\Phi}^{(r+s)}(0)}{z^2} + \frac{\widehat{\partial}^{(r)}\widehat{\Phi}^{(s)}(0)}{z} + \text{regular terms}\,, \tag{32}$$

where we have introduced the notations

$$\widehat{h}_\Phi^{(r)} = \sum_{j=1}^{N} e^{2i\pi rj/N} h_j\,, \quad \text{and} \quad \widehat{\partial}^{(r)} = \sum_{j=1}^{N} e^{2i\pi rj/N} (1 \otimes \ldots 1 \otimes \underset{(j-\text{th})}{\partial} \otimes 1 \otimes \ldots 1)\,. \tag{33}$$

This expression reduces to a simple form in the case of a principal primary operator:

$$\widehat{T}^{(r)}(z)\widehat{\phi}_h^{(s)}(0) = \frac{h\,\widehat{\phi}_h^{(r+s)}(0)}{z^2} + \frac{\partial\,\widehat{\phi}_h^{(r+s)}(0)}{z} + \text{regular terms}\,. \tag{34}$$

From the expression (32), the product of $\widehat{T}^{(r)}(z)$ with an untwisted primary operator is single-valued, and hence, only integer modes appear in the OPE:

$$\widehat{T}^{(r)}(z)\widehat{\Phi}^{(s)}(0) = \sum_{m\in\mathbb{Z}} z^{-m-2}\widehat{L}_m^{(r)}\widehat{\Phi}^{(s)}(0)\,. \tag{35}$$

**The twisted sectors**   The conical singularities of the surface $\Sigma_N$ are represented by twist operators in the orbifold theory. A twist operator of charge $k \neq 0$ is generically denoted as $\tau^{[k]}$, and corresponds to the end-point of a branch cut connecting the copies $j$ and $j+k$. If $A_j$ denotes the $j$-th copy of a given operator $A$ of dimension $h_A$, one has:

$$A_j(e^{2i\pi}z)\,\tau^{[k]}(0) = e^{-2i\pi h_A}A_{j+k}(z)\,\tau^{[k]}(0)\,. \tag{36}$$

This relation can be considered as a characterisation of an operator $\tau^{[k]}$ of the $k$-twisted sector.

As a consequence, the Fourier components $\widehat{T}^{(r)}$ have a simple monodromy around $\tau^{[k]}$:

$$\widehat{T}^{(r)}(e^{2i\pi}z)\,\tau^{[k]}(0) = e^{-2i\pi rk/N} \times \widehat{T}^{(r)}(z)\,\tau^{[k]}(0)\,, \tag{37}$$

and similarly for the primary operators $\widehat{\Phi}^{(r)}$ and $\widehat{\phi}_h^{(r)}$. Hence, the OPE of $\widehat{T}^{(r)}(z)$ with a twist operator can only include the modes consistent with this monodromy:

$$\widehat{T}^{(r)}(z)\,\tau^{[k]}(0) = \sum_{m \in \mathbb{Z}+kr/N} z^{-m-2}\,\widehat{L}_m^{(r)}\,\tau^{[k]}(0)\,. \tag{38}$$

If one supposes that there exists a "vacuum" operator $\tau_{\mathbb{1}}^{[k]}$ in the $k$-twisted sector, one can construct the other primary operators in this sector through the OPE:

$$\tau_\phi^{[k]}(z) := N^{h_\phi} \lim_{\epsilon \to 0} \left[ \epsilon^{(1-1/N)h_\phi}\,\tau_{\mathbb{1}}^{[k]}(z)\,(\phi(z+\epsilon) \otimes \mathbb{1} \otimes \cdots \otimes \mathbb{1}) \right]. \tag{39}$$

For convenience, in the following, we shall use the short-hand notations:

$$\tau_\phi := \tau_\phi^{[k=1]}\,, \qquad \widetilde{\tau}_\phi := \tau_\phi^{[k=-1]}\,. \tag{40}$$

In a sector of given twist, most fractional descendant act trivially:

$$L_{l/N}^{(r)}\tau^{[k]} = 0 \text{ if } l \notin N\mathbb{Z}+kr\,.$$

Hence the short-hand notation:

$$L_{l/N}\tau^{[k]} = 0 \equiv L_{l/N}^{(l \bmod N)/k}\tau^{[k]} = 0\,.$$

### 3.3   Induction procedure

Suppose one quantises the theory around a branch point of charge $k \neq 0$ at $z = 0$. After applying the conformal map $z \mapsto w = z^{1/N}$ from $\Sigma_N$ to a surface where $w = 0$ is a regular point, the currents $\widehat{T}^{(r)}(z)$ transform as:

$$\widehat{T}^{(r)}(z) \mapsto w^{2-2N}\sum_{j=0}^{N-1} e^{2i\pi j(r\ell+2)/N}\,T\left(e^{2i\pi j/N}w\right) + \frac{(N^2-1)c\,\delta_{r,0}}{24Nz^2}\,. \tag{41}$$

Accordingly, one gets for the generators:

$$\widehat{L}_m^{(r)} \mapsto \frac{1}{N}L_{Nm} + \frac{c}{24}\left(N-\frac{1}{N}\right)\delta_{r,0}\,\delta_{m,0}\,, \qquad \text{for} \quad m \in \mathbb{Z}+\frac{rk}{N}\,, \tag{42}$$

where the $L_n$'s are the ordinary Virasoro generators of the mother CFT. It is straightforward to check that these operators indeed obey the commutation relations (27). Similarly, the twisted operator $\tau_\phi^{[k]}$ (39) maps to the primary operator $\phi_h$ of the mother CFT:

$$\tau_\phi^{[k]}(z) \mapsto w^{(1-N)h_\phi}\,\phi(w)\,. \tag{43}$$

The relations (42–43) are called the "induction procedure" in [28]. Using (42–43) for $r = m = 0$, the dimension of $\tau_\phi^{[k]}$ for any $k \neq 0$ is

$$\widehat{h}_\phi = \frac{h_\phi}{N} + \frac{c}{24}\left(N - \frac{1}{N}\right). \tag{44}$$

This formula first appeared in [42, 43], and, in the context of entanglement entropy, in [44]. In particular, when $\phi$ is the identity operator $\mathbb{1}$ with dimension $h_\mathbb{1} = 0$, the expression (44) coincides with the dimension $h_\tau = \widehat{h}_\mathbb{1}$ (21) for $\tau_\mathbb{1}$.

## 3.4 Null-vector equations for untwisted and twisted operators

We intend to fully use the algebraic structure of the orbifold. If the mother theory is rational (i.e. it has a finite number of primary operators), then so is the orbifold theory. Also, from the induction procedure, we shall find null states for the twisted operators in the orbifold. In our approach, these null states are important, as they are the starting point of a conformal bootstrap approach.

**Non-twisted operators.** In the non-twisted sector, the null states are easy to compute. A state $\Phi$ in the non-twisted sector is a product of states of the mother theory. If one of these states (say, on the $j^{th}$ copy), has a null vector descendant in the mother theory, then the modes of $T_j(z)$ generate a null descendant. After an inverse discrete Fourier transform, these modes are easily expressed in terms of the orbifold Virasoro generators $\widehat{L}_n^{(r)}$.

For instance, take the mother CFT of central charge $c$, and consider the degenerate operator $\phi_{12}$. We can parameterise the central charge and degenerate conformal dimension as

$$c = 1 - \frac{6(1-g)^2}{g}, \qquad h_{12} = \frac{3g-2}{4}, \qquad 0 < g < 1, \tag{45}$$

and the null vector condition then reads:

$$\left(L_{-2} - \frac{1}{g}L_{-1}^2\right)\phi_{12} \equiv 0. \tag{46}$$

For a generic number of copies $N$ we have

$$\widehat{L}_n^{(r)} = \sum_{j=1}^N e^{2i\pi rj/N}(1 \otimes \ldots 1 \otimes \underset{(j-\text{th})}{L_n} \otimes 1 \otimes \ldots 1), \tag{47}$$

and hence for $N = 2$, and $\Phi = \phi_{12} \otimes \phi_{12}$, we obtain

$$\left[\widehat{L}_{-2}^{(0)} - \frac{1}{2g}\left(\widehat{L}_{-1}^{(0)}\right)^2 - \frac{1}{2g}\left(\widehat{L}_{-1}^{(1)}\right)^2\right]\Phi \equiv 0, \tag{48}$$

$$\left[\widehat{L}_{-2}^{(1)} - \frac{1}{g}\widehat{L}_{-1}^{(0)}\widehat{L}_{-1}^{(1)}\right]\Phi \equiv 0. \tag{49}$$

More generally, any product of degenerate operators from the mother CFT is itself degenerate under the orbifold Virasoro algebra.

**Twisted operators.** The twisted sectors also contain degenerate states, which are of great interest for the following. For example, let us take $k = 1$, and characterise the degenerate states at level $1/N$. A primary state $\tau$ obeys $\widehat{L}_m^{(r)}\tau = 0$ for any $r$ and positive $m \in \mathbb{Z} + r/N$. For $\tau$ to

be degenerate at level $1/N$, one needs to impose the additional constraint: $\widehat{L}^{(1)}_{1/N}\widehat{L}^{(-1)}_{-1/N}\tau = 0$. Using the commutation relations (27), we get:

$$\widehat{L}^{(1)}_{1/N}\widehat{L}^{(-1)}_{-1/N}\tau = \frac{2}{N}\left[\widehat{L}^{(0)}_0 - \frac{c}{24}\left(N - \frac{1}{N}\right)\right]\tau. \tag{50}$$

Thus, the operator $\tau$ is degenerate at level $1/N$ if and only if it has conformal dimension $h_\tau = \frac{c}{24}\left(N - \frac{1}{N}\right)$. This is nothing but the conformal dimension (21) of the vacuum twist operator $\tau_{\mathbb{1}}$. Hence, one always has

$$\widehat{L}^{(-1)}_{-1/N}\tau_{\mathbb{1}} \equiv 0, \qquad \widehat{L}^{(1)}_{-1/N}\widetilde{\tau}_{\mathbb{1}} \equiv 0. \tag{51}$$

A more generic method consists in using the induction procedure. First, (51) can be recovered by applying (42–43):

$$L_{-1}\mathbb{1} \equiv 0 \qquad \Rightarrow \qquad N\widehat{L}^{(-1)}_{-1/N}\tau_{\mathbb{1}} \equiv 0. \tag{52}$$

One can obtain the other twisted null-vector equations by the same induction principle. Let us give one more example in the $k = 1$ sector: the case of $\tau_{\phi_{12}}$. The relations (42–43) give:

$$\left(L_{-2} + g^{-1}L^2_{-1}\right)\phi_{12} \equiv 0 \qquad \Rightarrow \qquad \left[N\widehat{L}^{(-2)}_{-2/N} + \frac{N^2}{g}\left(\widehat{L}^{(-1)}_{-1/N}\right)^2\right]\tau_{\phi_{12}} \equiv 0. \tag{53}$$

and hence $\tau_{\phi_{12}}$ is degenerate at level $2/N$. Note the insertion of some factors $N$ in the null-vector of $\tau_{\phi_{12}}$ as compared to the null-vector equation (46) of the mother CFT.

# 4 First examples

## 4.1 Yang-Lee two-interval correlation function

We consider the CFT of the Yang-Lee (YL) singularity of central charge $c = -22/5$, where the primary operators are the identity $\mathbb{1} = \phi_{11} = \phi_{14}$ with conformal dimension $h_{\mathbb{1}} = 0$, and $\phi = \phi_{12} = \phi_{13}$ with $h_\phi = -1/5$. We shall compute the following four-point function in the $\mathbb{Z}_2$ orbifold of the YL model:

$$G(x, \bar{x}) = \langle\tau_{\mathbb{1}}(\infty)\tau_{\mathbb{1}}(1)\tau_{\mathbb{1}}(x, \bar{x})\tau_{\mathbb{1}}(0)\rangle. \tag{54}$$

In the $N = 2$ cyclic orbifold of the YL model, the untwisted primary operators are $\mathbb{1}$, $\Phi = \phi \otimes \phi$ and the principal primary fields $\widehat{\phi}^{(r)}$ with $r = 0, 1$. They have conformal dimensions, respectively, $h_{\mathbb{1}} = 0$, $h_\Phi = -2/5$ and $h_\phi = -1/5$. Note that for $N = 2$ the only twisted sector has $\ell = 1$, and hence $\widetilde{\tau} \equiv \tau$. For the same reason, we shall sometimes omit the superscripts on the generators $\widehat{L}^{(r)}_n$, as $r = 0, 1$ are the only possible values. The twisted primary operators are $\tau_{\mathbb{1}}$ and $\tau_\phi$, with conformal dimensions $\widehat{h}_{\mathbb{1}} = -11/40$ and $\widehat{h}_\phi = -3/8$. Here we have used the standard convention $\langle\psi(\infty)\dots\rangle := \lim_{R\to\infty}\left[R^{4h_\psi}\langle\psi(R)\dots\rangle\right]$. Geometrically, this correlation function correspond to the partition function of the Yang-Lee model on a twice branched sphere, which can be mapped to the torus.

The identity operator of the YL model satisfies two null-vector equations:

$$L_{-1}\mathbb{1} = 0, \qquad \left(L_{-4} - \frac{5}{3}L^2_{-2}\right)\mathbb{1} = 0. \tag{55}$$

Through the induction procedure, this yields null-vector equations for the twist operator $\tau_{\mathbb{1}}$:

$$\widehat{L}_{-1/2}\tau_{\mathbb{1}} = 0, \qquad \left(\widehat{L}_{-2} - \frac{10}{3}\widehat{L}_{-1}^2\right)\tau_{\mathbb{1}} = 0, \tag{56}$$

where the Fourier modes are $r = 1$ and $r = 0$ respectively, as required from (38). The first equation of (56) is generic for all $N = 2$ orbifolds, and determines the conformal dimension of the $\tau_{\mathbb{1}}$ operator. In contrast, the second equation is specific to the YL model. It only involves the integer modes, which all have the usual differential action when inserted into a correlation function. Hence, due to the second equation of (56), the derivation of $G(x, \bar{x})$ is very similar to the standard case of a four-point function involving the degenerate operator $\phi_{12}$ (see appendix B). The conformal block in $z \to 0$ have the expression:

$$x^{11/20}(1-x)^{11/20}I_1(x) \text{ with } I_1(x) = {}_2F_1(7/10, 11/10; 7/5|x),$$
$$x^{11/20}(1-x)^{11/20}I_2(x) \text{ with } I_2(x) = x^{-\frac{2}{5}}{}_2F_1(7/10, 3/10; 3/5|x), \tag{57}$$

and the total correlation function can be written:

$$G(x, \bar{x}) = |x|^{11/10}|1-x|^{11/10} \times \Big[ \left|{}_2F_1(7/10, 11/10; 7/5|x)\right|^2$$
$$+ 2^{16/5}\left|x^{-2/5}{}_2F_1(7/10, 3/10; 3/5|x)\right|^2 \Big], \tag{58}$$

**OPE coefficients**   The coefficients $X_j$ and $Y_j$ give access to the OPE coefficients in the $\mathbb{Z}_2$ orbifold of the YL model:

$$C(\tau_{\mathbb{1}}, \tau_{\mathbb{1}}, \Phi) = \sqrt{X_2} = 2^{8/5}. \tag{59}$$

Recalling $h_\phi = -1/5$, we see that (188) is consistent with the expression $C(\Phi, \tau_{\mathbb{1}}, \tau_{\mathbb{1}}) = 2^{-8h_\phi}$ (see Appendix C).

**Mapping to the torus**   The mapping from the torus (with coordinates $t$) to the branched sphere (with coordinates $z$) is:

$$z(t) = \frac{\wp(t) - \wp(1/2)}{\wp(\tau/2) - \wp(1/2)}. \tag{60}$$

This maps $0, x, 1, \infty \leftarrow \frac{1}{2}, \frac{1}{2}(1+\tau), \frac{\tau}{2}, 0$. The relation between $x$ and the nome $q = e^{2i\pi\tau}$ is given by:

$$x = 16\sqrt{q}\prod_{n=1}^{\infty}\left(\frac{1+q^n}{1+q^{n-1/2}}\right)^8. \tag{61}$$

Mapping the torus to the branched sphere, the partition function transforms as:

$$Z(\tau) = 4^{c/3}|x|^{c/12}|1-x|^{c/12}\langle\tau_{\mathbb{1}}|\tau_{\mathbb{1}}(1)\tau_{\mathbb{1}}(x, \bar{x})|\tau_{\mathbb{1}}\rangle. \tag{62}$$

The torus partition function of the Yang-Lee model involves two characters, $\chi_{1,1}(\tau)$ ($\mathbb{1}$), and $\chi_{1,2}(\tau)$ ($\phi$).

$$Z = |\chi_{3,1}(\tau)|^2 + |\chi_{4,1}(\tau)|^2. \tag{63}$$

The characters of minimal models have the well-known expression $\chi_{r,s}(\tau) = K_{r,s}(\tau) - K_{r,-s}(\tau)$, with:

$$K_{r,s}(\tau) = \frac{1}{\eta(\tau)}\sum_{n=-\infty}^{\infty}\exp\left(i\pi\tau\frac{(20n+2r-5s)^2}{20}\right), \tag{64}$$

where $\eta$ is the Dedekind eta function.

We expect the following relation between the conformal blocks of the correlation function and the characters of the theory:

$$\chi_{1,1}(\tau) = 2^{-22/15} x^{11/30} (1-x)^{11/30} I_1(x), \qquad \chi_{1,2}(\tau) = 2^{2/15} x^{11/30} (1-x)^{11/30} I_2(x).$$

Those two relations are not trivial, and the simplest way to prove them seems to be by showing that the right-hand terms are vector modular form, with the same modular transformations as $\chi$ (see by example [45] for details). An easy check consist in expanding the right-hand side in power of $q$, confirming the equality for first orders:

$$2^{-22/15} x^{11/30} (1-x)^{11/30} I_1(x) \approx 1 + q^2 + q^3 + q^4 + \cdots$$

$$2^{2/15} x^{11/30} (1-x)^{11/30} I_2(x) \approx 1 + q + q^2 + q^3 + 2q^4 + \cdots$$

The relation 62 between the partition on the torus and $G(x,\bar{x})$ is verified:

$$Z(\tau) = 2^{-44/15} |x|^{-11/30} |1-x|^{-11/30} \langle \tau_{\mathbb{1}} | \tau_{\mathbb{1}}(1) \tau_{\mathbb{1}}(x,\bar{x}) | \tau_{\mathbb{1}} \rangle. \tag{65}$$

## 4.2 Yang-Lee one-interval correlation function

With minimal modifications to the previous argument, we can also compute the following four-point function :

$$G(x,\bar{x}) = \langle \Phi(\infty) \tau_{\mathbb{1}}(1) \tau_{\mathbb{1}}(x,\bar{x}) \Phi(0) \rangle. \tag{66}$$

Which, physically, is related to the generalised Rényi entropy $S_{N=2}(x,\phi,\tau_{\mathbb{1}})$.

Technically, it is more convenient to work with twist operators located at 0 and $\infty$, so we introduce

$$F(x,\bar{x}) := \langle \tau_{\mathbb{1}}(\infty) \Phi(1) \Phi(x,\bar{x}) \tau_{\mathbb{1}}(0) \rangle. \tag{67}$$

Using the suitable projective mapping, one has the relation:

$$G(x,\bar{x}) = |1-x|^{4(h_\Phi - \widehat{h}_{\mathbb{1}})} F(x,\bar{x}).$$

Then, through the null-vector of $\tau_{\mathbb{1}}$, we can obtain a differential equation of order two for this correlation function:

$$\left[ 10x^2(1-x)^2 \partial_x^2 + x(1-x)(3-x)\partial_x + \frac{2}{5}(5x^2+3) \right] F(x,\bar{x}) = 0. \tag{68}$$

The Riemann scheme of this equation is:

$$
\begin{array}{ccc}
0 & 1 & \infty \\
\hline
\frac{2}{5} & \frac{4}{5} & -\frac{2}{5} \\
\frac{3}{10} & \frac{2}{5} & -\frac{1}{2}
\end{array}, \tag{69}
$$

which is consistent with the OPEs:

$$\Phi \times \tau_{\mathbb{1}} \to \tau_{\mathbb{1}} + \tau_\phi, \tag{70}$$

$$\Phi \times \Phi \to \mathbb{1} + \Phi + (1 \otimes \phi), \tag{71}$$

$$\tau_{\mathbb{1}} \times \tau_{\mathbb{1}} \to \mathbb{1} + \Phi. \tag{72}$$

By appropriately shifting the function $F$, $F(x,\bar{x}) = |x|^{4/5} |1-x|^{8/5} f(x,\bar{x})$, we can turn 68 in 174. At that point we can simply re-use the results of the last sections with parameters:

$$a = \frac{4}{5}, \qquad b = \frac{7}{10}, \qquad c = \frac{11}{10}. \tag{73}$$

The final result for the four-point function $G(x, \bar{x})$ (66) is

$$
\begin{aligned}
G(x, \bar{x}) = |x|^{4/5} |1-x|^{11/5} \times \Big[ & X_1 \left| {}_2F_1(4/5, 7/10; 11/10|x) \right|^2 \\
& + X_2 \left| x^{-1/10} {}_2F_1(3/5, 7/10; 9/10|x) \right|^2 \Big],
\end{aligned}
\tag{74}
$$

where $X_1, X_2$ are given in (188), and the parameters $a, b, c, d$ are given in (73). Using the identity (166) on hypergeometric functions, we see that the solution (74) for $G(x, \bar{x})$ agrees with the direct computation given in Appendix D.

## 5 Twist operators with a fractional null vector

### 5.1 Orbifold Ward identities

Generically the null vectors for a twist operator can involve some generators $L_m^{(r)}$, with $r \neq 1$ and fractional indices $m \in \mathbb{Z} + r/N$ [see (38)], which do not have a differential action on the correlation function. In this situation, we shall use the extended Ward identities to turn the null-vector conditions into a differential equation for the correlation function.

Let us consider the correlation function:

$$
\mathscr{G}^{(r)}(x, \bar{x}, z) = \langle \mathscr{O}_1 | \mathscr{O}_2(1) \mathscr{O}_3(x, \bar{x}) \widehat{T}^{(r)}(z) | \mathscr{O}_4 \rangle,
\tag{75}
$$

where $(\mathscr{O}_2, \mathscr{O}_3)$ are any two operators and $(|\mathscr{O}_1\rangle, |\mathscr{O}_4\rangle)$ are any two states of the cyclic orbifold. Each operator $\mathscr{O}_j$ or state $|\mathscr{O}_j\rangle$ can be in a twisted sector $[k_j]$ with $k_j \neq 0 \mod N$, or in the untwisted sector ($k_j \equiv 0 \mod N$). The overall $\mathbb{Z}_N$ symmetry imposes a neutrality condition: $k_1 + k_2 + k_3 + k_4 \equiv 0 \mod N$. Let $C$ be a contour enclosing the points $\{0, x, 1\}$. Then this is a closed contour for the following integral:

$$
\frac{1}{2i\pi} \oint_C dz \, (z-1)^{m_2+1}(z-x)^{m_3+1} z^{m_4+1} \mathscr{G}^{(r)}(x, \bar{x}, z),
\tag{76}
$$

where $m_j \in \mathbb{Z} + r k_j/N$ for $j = 2, 3, 4$. Then, by deforming the integration contour to infinity, we obtain the following identity:

$$
\begin{aligned}
\sum_{p=0}^{\infty} a_p \langle \mathscr{O}_1 | \widehat{L}_{-m_1-p}^{(r)} \mathscr{O}_2(1) \mathscr{O}_3(x, \bar{x}) | \mathscr{O}_4 \rangle = & \sum_{p=0}^{\infty} b_p \langle \mathscr{O}_1 | [\widehat{L}_{m_2+p}^{(r)} \mathscr{O}_2](1) \mathscr{O}_3(x, \bar{x}) | \mathscr{O}_4 \rangle \\
& + \sum_{p=0}^{\infty} c_p \langle \mathscr{O}_1 | \mathscr{O}_2(1) [\widehat{L}_{m_3+p}^{(r)} \mathscr{O}_3](x, \bar{x}) | \mathscr{O}_4 \rangle \\
& + \sum_{p=0}^{\infty} d_p \langle \mathscr{O}_1 | \mathscr{O}_2(1) \mathscr{O}_3(x, \bar{x}) \widehat{L}_{m_4+p}^{(r)} | \mathscr{O}_4 \rangle,
\end{aligned}
\tag{77}
$$

where

$$
m_1 = -m_2 - m_3 - m_4 - 2 \quad \Rightarrow \quad m_1 \in \mathbb{Z} + r k_1/N,
\tag{78}
$$

and the coefficients $a_p, b_p, c_p, d_p$ are defined by the Taylor expansions:

$$
\begin{aligned}
(1-z)^{m_2+1}(1-xz)^{m_3+1} = \sum_{p=0}^{\infty} a_p z^p, \qquad & (z-x)^{m_3+1} z^{m_4+1} = \sum_{p=0}^{\infty} b_p (z-1)^p, \\
(z-1)^{m_2+1} z^{m_4+1} = \sum_{p=0}^{\infty} c_p (z-x)^p, \qquad & (z-1)^{m_2+1}(z-x)^{m_3+1} = \sum_{p=0}^{\infty} d_p z^p.
\end{aligned}
\tag{79}
$$

In (77) we have used the notation:

$$[\widehat{L}_m^{(r)}\mathcal{O}_j](x,\bar{x}) := \frac{1}{2i\pi}\oint_{C_x} dz\,(z-x)^{m+1}\,\widehat{T}^{(r)}(z)\mathcal{O}_j(x,\bar{x}),\tag{80}$$

where $C_x$ is a contour enclosing the point $x$. If all the $\mathcal{O}_j$'s are chosen among the primary operators or their descendants under the orbifold Virasoro algebra, then the sums in (77) become finite. By choosing appropriately the indices $m_1,\dots m_4$ (with the constraint $\sum_j m_j = -2$), the relations (77) can be used to express, say, $\langle \widehat{L}_{r/N}^{(r)}\tau\dots\rangle$ in terms of correlation functions involving only descendants with integer indices.

## 5.2 Ising two-interval ground state entropy

The Ising model is the smallest non-trivial unitary model, with three fields $\{\mathbb{1},\sigma,\epsilon\}$, of conformal charges $0,1/16,1/2$. Its central charge is $\frac{1}{2}$. By using the correspondance between the Ising model and free fermions, the two-interval case was computed in [16], for any value of $N$. Here, as an example, we compute the $N=2$ two-interval entropy, with our method, and using only the null vector of the model. The $N=2$ orbifold of the Ising model will have central charge 1. The first field in the twisted sector is $\tau_{\mathbb{1}}$, the identity twist, of charge $h_{\tau_{\mathbb{1}}} = 1/32$.

The two null vector of the identity in the mother theory are:

$$L_{-1}\mathbb{1} = 0,\qquad \left(108L_{-6} + 264L_{-4}L_{-2} - 93\,L_{-3}^2 - 64\,L_{-2}^3\right)\mathbb{1} = 0.\tag{81}$$

And, through the induction process, we obtain null vector equations for $\tau_{\mathbb{1}}$.

$$L_{-1/2}\tau_{\mathbb{1}} = 0\qquad \left(54\,L_{-3} + 264\,L_{-2}L_{-1} - 93\,L_{-3/2}^2 - 128\,L_{-1}^3\right)\tau_{\mathbb{1}} = 0.\tag{82}$$

We aim to obtain a differential equation for the correlation function

$$G(x,\bar{x}) = \langle\tau_{\mathbb{1}}|\tau_{\mathbb{1}}(x,\bar{x})\tau_{\mathbb{1}}(1)|\tau_{\mathbb{1}}\rangle,$$

We need to compute the term $\langle\left(L_{-3/2}^2\tau_{\mathbb{1}}(0)\right)\tau_{\mathbb{1}}(x,\bar{x})\tau_{\mathbb{1}}(1)\tau_{\mathbb{1}}(\infty)\rangle$ to be able to use the null-vector. By using the Ward identity 77 with $\{m_1,m_2,m_3,m_4\} = \{1/2,-1/2,-1/2,-3/2\}$ and $\{\mathcal{O}_1,\mathcal{O}_2,\mathcal{O}_3,\mathcal{O}_4\} = \{\tau_{\mathbb{1}},\tau_{\mathbb{1}},\tau_{\mathbb{1}},L_{-3/2}\tau_{\mathbb{1}}\}$, we obtain the relation:

$$\sum_{p=0}^{3} d_p\langle\tau_{\mathbb{1}}|\tau_{\mathbb{1}}(1)\tau_{\mathbb{1}}(x,\bar{x})L_{-3/2+p}|L_{-3/2}\tau_{\mathbb{1}}\rangle = 0,$$

which, using the orbifold algebra is equivalent to:

$$\langle\tau_{\mathbb{1}}|\tau_{\mathbb{1}}(x,\bar{x})\tau_{\mathbb{1}}(1)|L_{-3/2}^2\tau_{\mathbb{1}}\rangle = \frac{1}{64x^3}\Big(32x^2(1+x)\langle\tau_{\mathbb{1}}|\tau_{\mathbb{1}}(x,\bar{x})\tau_{\mathbb{1}}(1)|L_{-2}\tau_{\mathbb{1}}\rangle$$
$$+16x(x-1)^2\langle\tau_{\mathbb{1}}|\tau_{\mathbb{1}}(x,\bar{x})\tau_{\mathbb{1}}(1)|L_{-1}\tau_{\mathbb{1}}\rangle + (x-1)^2(1+x)\langle\tau_{\mathbb{1}}|\tau_{\mathbb{1}}(x,\bar{x})\tau_{\mathbb{1}}(1)|\tau_{\mathbb{1}}\rangle\Big).$$

Using this relation and the null vector 82, we obtain the following differential equation:

$$\Big(15(2x-1) + 48x(x-1)(192x^2 - 192x + 5)\partial_x$$
$$+ 8448(x-1)^2x^2(2x-1)\partial_x^2 + 4096(x-1)^3x^3\partial_x^3\Big)G(x,\bar{x}) = 0.\tag{83}$$

The Riemann scheme of this equation is:

$$
\begin{array}{ccc}
0 & 1 & \infty \\
\hline
\frac{-1}{16} & \frac{1}{16} & \frac{15}{16} \\
\frac{-1}{16} & \frac{1}{16} & \frac{15}{16} \\
0 & -\frac{1}{8} & -1
\end{array}
\,,
\tag{84}
$$

which is compatible with the OPE:

$$
\tau_{\mathbb{1}} \otimes \tau_{\mathbb{1}} \to \mathbb{1}^{\otimes 2} \oplus \sigma^{\otimes 2} \oplus \epsilon^{\otimes 2}\,.
$$

Under the change of variables $C(x,\bar{x}) = |x|^{-1/24}|1-x|^{-1/24}G(x,\bar{x})$, the differential equation 83 becomes:

$$
\left(\partial_x^3 - \frac{(2(2x-1))\partial_x^2}{x(1-x)} + \frac{\left(391\left(x^2-x\right)+7\right)\partial_x}{192x^2(1-x)^2} \right.
$$
$$
\left. - \frac{23(2-x)(x+1)(2x-1)}{24^3 x^3(1-x)^3}\right)C(x,\bar{x}) = 0\,.
\tag{85}
$$

The solution of this differential equation are the characters of the Ising model on the torus, as demonstrated in [46], by directly applying the null vectors on the torus. Hence we recover that the two-interval $N=2$ Rényi entropy maps to the torus partition function.

## 5.3 Yang-Lee one-interval ground state entropy

**Definitions.**    As argued above, the $N=2$ ground state entropy of the YL model is related to the correlation function:

$$
G(x,\bar{x}) = \langle \Phi(\infty)\tau_\phi(1)\tau_\phi(x,\bar{x})\Phi(0)\rangle\,,
\tag{86}
$$

where $\Phi = \phi \otimes \phi$. It will be convenient to work rather with the related function:

$$
F(x,\bar{x}) = \langle \tau_\phi(\infty)\Phi(1)\Phi(x,\bar{x})\tau_\phi(0)\rangle\,,
\tag{87}
$$

with $G(x,\bar{x}) = |1-x|^{4(h_\Phi - \widehat{h}_\phi)}F(x,\bar{x})$.

**Null vectors and independent descendants.**    In the mother theory, the primary field $\phi$ has two null-descendants:

$$
\left(L_{-2} - \frac{5}{2}L_{-1}^2\right)\phi = 0\,, \qquad \left(L_{-3} - \frac{25}{12}\widehat{L}_{-1}^3\right)\phi = 0\,.
\tag{88}
$$

Through the induction procedure, this implies the following null vectors for the twist field $\tau_\phi$:

$$
\left(\widehat{L}_{-1} - 5\,\widehat{L}_{-1/2}^2\right)\tau_\phi = 0\,, \qquad \left(\widehat{L}_{-3/2} - \frac{25}{4}\,\widehat{L}_{-1/2}^3\right)\tau_\phi = 0\,.
\tag{89}
$$

The Ward identities (77) will give relations between descendants at different levels. In order to get an idea of the descendants we need to compute, we need to know the number of independent descendants at each level. The number of Virasoro descendants at a given level $k$ is equal to the number of integer partitions of $k$. In a minimal model not all those descendants are independent. The number of (linearly) independent descendants at level $k$ of a primary field $\phi$ is given by the coefficients of the series expansion of the character associated with $\phi$.

Explicitly, if $\tau$ is the modular parameter on the torus, the character of a field $\phi_{rs}$ in the minimal model $\mathscr{M}_{p,p'}$ is given by:

$$\chi_{rs}(\tau) = K_{pr-p's}(\tau) - K_{pr+p's}(\tau), \qquad K_\lambda(\tau) = \frac{1}{\eta(\tau)} \sum_{n\in\mathbb{Z}} q^{(2pp'n+\lambda)^2/2pp'}, \qquad q = e^{2i\pi\tau}.$$

The coefficient of order $k$ in the series expansion with parameter $q$ of the character gives the number of independent fields at level $k$. Through the induction procedure (42–43), the module of $\tau_{\phi_{rs}}$ under the orbifold algebra has the same structure as the module of $\phi_{rs}$ under the Virasoro algebra. Hence this coefficient also gives the number of descendants at level $k/N$ in the module of $\tau_{\phi_{rs}}$. For example, for $N = 2$ and $(p, p') = (5, 2)$ (Yang-Lee), the numbers of independent descendants for the field $\tau_\phi = \tau_{\phi_{12}}$ are given in the following table.

| level | 1/2 | 1 | 3/2 | 2 | 5/2 | 3 |
|---|---|---|---|---|---|---|
| descendants | 1 | 1 | 1 | 2 | 2 | 3 |
| integer descendants | 0 | 1 | 0 | 2 | 0 | 3 |

So up to level 3, all descendants at an integer level (even formed of non-integer descendants) can be re-expressed in terms of integer modes. One gets explicitly:

$$\widehat{L}_{-5/2}\widehat{L}_{-1/2}\tau_\phi = \left(\frac{1}{2}\widehat{L}_{-3} + \frac{3}{2}\widehat{L}_{-1}\widehat{L}_{-2} - \frac{5}{3}\widehat{L}_{-1}^3\right)\tau_\phi,$$

$$\widehat{L}_{-3/2}\widehat{L}_{-1/2}\tau_\phi = \left(\frac{1}{4}\widehat{L}_{-2} - \frac{1}{2}\widehat{L}_{-1}^2\right)\tau_\phi, \tag{90}$$

$$\widehat{L}_{-1/2}^2\tau_\phi = \frac{1}{5}\widehat{L}_{-1}\tau_\phi, \qquad \widehat{L}_{1/2}\widehat{L}_{-1/2}\tau_\phi = \left(\widehat{L}_0 - \frac{c}{16}\right)\tau_\phi.$$

**Ward identity.**   We now use (77) with the indices $(m_1, \ldots, m_4) = (1/2, 0, 0, -5/2)$ and:

$$\mathscr{G}(x, \bar{x}, z) = \langle\tau_\phi|\Phi(1)\Phi(x,\bar{x})\widehat{T}^{(1)}(z)\widehat{L}_{-1/2}^{(1)}|\tau_\phi\rangle.$$

Recalling that $\widehat{L}_0^{(1)}\Phi = 0$, the only surviving terms are:

$$0 = \sum_{p=0}^\infty d_p \langle\tau_\phi|\Phi(1)\Phi(x,\bar{x})\widehat{L}_{-5/2+p}^{(1)}\widehat{L}_{-1/2}^{(1)}|\tau_\phi\rangle.$$

Inserting the explicit expressions for the coefficients $d_p$, we get:

$$\langle\tau_\phi|\Phi(1)\Phi(x,\bar{x})\left[x\widehat{L}_{-5/2} - (x+1)\widehat{L}_{-3/2} + \widehat{L}_{-1/2}\right]\widehat{L}_{-1/2}|\tau_\phi\rangle = 0. \tag{91}$$

**Differential equation.**   Combining (90) and (91), one gets a linear relation involving only the integer modes $\widehat{L}_m^{(0)}$. Using (169), this leads to the third-order differential equation:

$$\left[\frac{5}{3}x^3(1-x)^3\partial_x^3 + 2x^2(1-x)^2(1-2x)\partial_x^2 \right.$$

$$\left. + \frac{1}{20}x(1-x)(15x^2 - 14x + 7)\partial_x - \frac{1}{50}(x^3 - 3x^2 - 29x + 15)\right]F(x,\bar{x}) = 0. \tag{92}$$

The Riemann scheme of this equation is:

$$\begin{array}{ccc} 0 & 1 & \infty \\ \hline \frac{1}{2} & \frac{4}{5} & \frac{1}{10} \\ \frac{2}{5} & \frac{2}{5} & -\frac{3}{10} \\ \frac{9}{10} & \frac{3}{5} & -\frac{2}{5} \end{array} \tag{93}$$

**Interpretation in terms of OPEs.** We consider the conformal blocks under the invariant subalgebra $\mathscr{A}_0$ generated by the monomials $L_{m_1}^{(r_1)} \dots L_{m_k}^{(r_k)}$ with $r_1 + \cdots + r_k \equiv 0 \mod N$. With this choice, the toroidal partition function of the orbifold has a diagonal form $Z = \sum_j |\chi_j|^2$ in terms of the characters (see [27,28]). The OPEs under the invariant subalgebra $\mathscr{A}_0$ are then:

$$\Phi \times \tau_\phi \to \tau_{\mathbb{1}} + \tau_\phi + \widehat{L}_{-1/2}^{(1)} \tau_\phi \,, \tag{94}$$

$$\Phi \times \Phi \to \mathbb{1} + \Phi + (1 \otimes \phi) \,, \tag{95}$$

$$\tau_\phi \times \tau_\phi \to \mathbb{1} + \Phi + (1 \otimes \phi) \,. \tag{96}$$

Note that $\widehat{L}_{-1/2}^{(1)} \tau_\phi$ is a primary operator with respect to $\mathscr{A}_0$. The local exponents (93) are consistent with these OPEs.

**Holomorphic solutions.** To express the solutions in terms of power series around $x = 0$, it is convenient to rewrite (92) using the differential operator $\theta = x \partial_x$. On has, for any $k \in \mathbb{N}$:

$$x^k \partial_x^k = \theta(\theta - 1) \dots (\theta - k + 1) \,.$$

This yields the new form for (92):

$$\left[ P_0(\theta) + x P_1(\theta) + x^2 P_2(\theta) + x^3 P_3(\theta) \right] F(x, \bar{x}) = 0 \,, \tag{97}$$

where:

$$
\begin{aligned}
P_0(\theta) &= \frac{1}{3}(2\theta - 1)(5\theta - 2)(10\theta - 9) \,, \\
P_1(\theta) &= -\frac{1}{5}(500\theta^3 - 700\theta^2 + 305\theta - 58) \,, \\
P_2(\theta) &= \frac{1}{5}(500\theta^3 - 500\theta^2 + 145\theta + 6) \,, \\
P_3(\theta) &= -\frac{1}{15}(5\theta - 2)(10\theta - 3)(10\theta + 1) \,.
\end{aligned}
\tag{98}
$$

The key identity satisfied by the operator $\theta$ is, for any polynomial $P$ and any real $\alpha$:

$$P(\theta) . x^\alpha = P(\alpha) x^\alpha \,. \tag{99}$$

Hence, if we choose $\alpha$ to be a root of $P_0$ (i.e. one of the local exponents at $x = 0$), there exists exactly one solution of the form:

$$I(x) = x^\alpha \times \sum_{n=0}^{\infty} a_n x^n \,, \qquad \text{with } a_n = 1 \,. \tag{100}$$

The coefficients $a_n$ are given by the linear recursion relation:

$$a_n = -\frac{P_1(\alpha + n) a_{n-1} + P_2(\alpha + n) a_{n-2} + P_3(\alpha + n) a_{n-3}}{P_0(\alpha + n)} \,, \tag{101}$$

with the initial conditions:

$$a_0 = 1 \,, \qquad a_1 = -\frac{P_1(\alpha + 1)}{P_0(\alpha + 1)} \,, \qquad a_2 = \frac{P_1(\alpha + 1) P_1(\alpha + 2) - P_0(\alpha + 1) P_2(\alpha + 2)}{P_0(\alpha + 1) P_0(\alpha + 2)} \,. \tag{102}$$

For example, the conformal block corresponding to $\alpha = 1/2$:

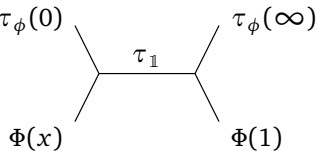

is given by

$$I_1(x) = x^{1/2}\left(1 + \frac{256}{55}x + \frac{24446}{1925}x^2 + \dots\right).$$ (103)

The series converges for $|x| < 1$, and may be evaluated numerically to arbitrary precision using (101).

**Numerical solution of the monodromy problem.** To determine the physical correlation function $F(x, \bar{x})$ (87), we need to solve the monodromy problem, *i.e.* find the coefficients for the decompositions:

$$F(x, \bar{x}) = \sum_{i=1}^{3} X_i |I_i(x)|^2 = \sum_{j=1}^{3} Y_j |J_j(x)|^2,$$ (104)

where $(I_1, I_2, I_3)$ are the holomorphic solutions of (92) with exponents $(1/2, 2/5, 9/10)$ around $x = 0$, and $(J_1, J_2, J_3)$ are the holomorphic solutions with exponents $(4/5, 2/5, 3/5)$ around $x = 1$. In the present case, we do not know the analytic form of the $3 \times 3$ matrix $A$ for the change of basis:

$$I_i(x) = \sum_{j=1}^{3} A_{ij} J_j(x).$$ (105)

However, this matrix can be evaluated numerically with arbitrary precision, by replacing the $I_i$'s and $J_j$'s in (105) by their series expansions of the form (103), and generating a linear system for $\{A_{ij}\}$ by choosing any set of points $\{x_1, \dots, x_9\}$ on the interval $0 < x < 1$ where both sets of series $I_i(x)$ and $J_j(x)$ converge. We obtain the matrix:

$$A = \begin{pmatrix} 0.46872 & 2.98127 & -2.61803 \\ 0.292217 & 2.43298 & -1.82483 \\ 3.52145 & 6.92136 & -9.83452 \end{pmatrix}.$$ (106)

The coefficients giving a monodromy-invariant $F(x, \bar{x})$ consistent with the change of basis (105) are:

$$\begin{aligned} X_1 &= 30.6594, \\ X_2 &= -19.2813, \\ X_3 &= 0.211121, \\ Y_1 &= 1, \\ Y_2 &= 20.2276, \\ Y_3 &= -9.64063. \end{aligned}$$ (107)

We have set $Y_1$ to one because it corresponds to the conformal block with the identity as the internal operator.

**OPE coefficients.** The coefficients $X_i$ and $Y_j$ are related to the OPE coefficients in the $N = 2$ orbifold of the YL model as follows:

$$X_1 = C(\Phi, \tau_\phi, \tau_\mathbb{1})^2 , \tag{108}$$

$$X_2 = C(\Phi, \tau_\phi, \tau_\phi)^2 , \tag{109}$$

$$X_3 = C(\Phi, \tau_\phi, \widehat{L}^{(1)}_{-1/2} \tau_\phi)^2 , \tag{110}$$

$$Y_1 = C(\Phi, \Phi, \mathbb{1}) C(\mathbb{1}, \tau_\phi, \tau_\phi) = 1 , \tag{111}$$

$$Y_2 = C(\Phi, \Phi, \Phi) C(\Phi, \tau_\phi, \tau_\phi) , \tag{112}$$

$$Y_3 = C[\Phi, \Phi, (\phi \otimes \mathbb{1})] C[(\phi \otimes \mathbb{1}), \tau_\phi, \tau_\phi] . \tag{113}$$

Our numerical procedure can be checked by comparing some of the numerical values (107) to a direct calculation of the OPE coefficients done in Appendix C. They match up to machine precision.

## 5.4 Excited state entropy for minimal models at $N = 2$

The same type of strategy can be used to compute excited state entropies in other minimal models, for some specific degenerate states, and small values of $N$. In this section we solve explicitly the case of $N = 2$ for an operator degenerate at level 2.

**Definitions.** We consider the minimal model $\mathcal{M}(p, p')$ with central charge $c = 1 - \frac{6(p-p')^2}{pp'}$, where $p, p'$ are coprime integers. Using the Coulomb-gas notation, $\phi_{21}$ is one of the states of the mother CFT which possess a null vector at level 2. In terms of the Coulomb-gas parameter $g = p/p'$, the central charge and conformal dimensions of the Kac table read:

$$c = 1 - \frac{6(1-g)^2}{g} , \tag{114}$$

$$h_{rs} = \frac{(rg-s)^2 - (1-g)^2}{4g} , \qquad \text{for} \quad \begin{cases} r = 1, \ldots, p'-1 \\ s = 1, \ldots, p-1 . \end{cases} \tag{115}$$

In particular, we have $h_{21} = (3g-2)/4$. The null-vector condition for $\phi_{21}$ reads:

$$\left( L_{-2} - \frac{1}{g} L_{-1}^2 \right) \phi_{21} = 0 . \tag{116}$$

As shown before, in a unitary model (when $q = p-1$), the entanglement entropy of an interval in the state $|\phi_{21}\rangle$ is expressed in terms of the correlation function

$$G(x, \bar{x}) = \langle \Phi(\infty) \tau_\mathbb{1}(1) \tau_\mathbb{1}(x, \bar{x}) \Phi(0) \rangle , \tag{117}$$

where $\Phi = \phi_{21} \otimes \phi_{21}$.

**Null vectors.** The null vectors of $\tau_\mathbb{1}$ and $\Phi$ can be obtained through the induction procedure:

$$\widehat{L}^{(1)}_{-1/2} \tau_\mathbb{1} = 0 , \tag{118}$$

$$\left[ \widehat{L}^{(0)}_{-2} - \frac{1}{2g} \left( (\widehat{L}^{(0)}_{-1})^2 + (\widehat{L}^{(1)}_{-1})^2 \right) \right] \Phi = 0 , \qquad \left( \widehat{L}^{(1)}_{-2} - \frac{1}{g} \widehat{L}^{(0)}_{-1} \widehat{L}^{(1)}_{-1} \right) \Phi = 0 . \tag{119}$$

**Ward identity.** Using the Ward identity (77) with the choice of indices $(m_1, \cdots, m_4) = (0, -1/2, -1/2, -1)$ for the function

$$\mathscr{G}(x, \bar{x}, z) = \langle \Phi | \tau_\phi(1) \tau_\phi(x, \bar{x}) \widehat{T}^{(1)}(z) \widehat{L}^{(1)}_{-1} | \Phi \rangle,$$

and the null-vector condition (118), we obtain the relation:

$$0 = \langle \Phi | \tau_{\mathbb{1}}(1) \tau_{\mathbb{1}}(x, \bar{x}) \left( d_0 \widehat{L}^{(1)}_{-1} + d_1 \widehat{L}^{(1)}_0 + d_2 \widehat{L}^{(1)}_1 \right) \widehat{L}^{(1)}_{-1} | \Phi \rangle. \tag{120}$$

Using the orbifold Virasoro algebra (27) and the explicit expression for the $d_p$'s, we get:

$$\langle \Phi | \tau_{\mathbb{1}}(1) \tau_{\mathbb{1}}(x, \bar{x}) (\widehat{L}^{(1)}_{-1})^2 | \Phi \rangle = \frac{x+1}{2x} \langle \Phi | \tau_{\mathbb{1}}(1) \tau_{\mathbb{1}}(x, \bar{x}) \widehat{L}^{(0)}_{-1} | \Phi \rangle$$

$$+ \frac{h_{21}(x-1)^2}{2x^2} \langle \Phi | \tau_{\mathbb{1}}(1) \tau_{\mathbb{1}}(x, \bar{x}) | \Phi \rangle. \tag{121}$$

**Differential equation.** By inserting the first null-vector condition of (119) into $G(x, \bar{x})$, we get:

$$\langle \Phi | \tau_{\mathbb{1}}(1) \tau_{\mathbb{1}}(x, \bar{x}) \left[ \widehat{L}^{(0)}_{-2} - \frac{1}{2g} \left( (\widehat{L}^{(0)}_{-1})^2 + (\widehat{L}^{(1)}_{-1})^2 \right) \right] | \Phi \rangle = 0. \tag{122}$$

Then, the substitution of the $(\widehat{L}^{(1)}_{-1})^2$ term by (121) gives a linear relation involving only the modes $\widehat{L}^{(0)}_m$, which have a differential action on $G(x, \bar{x})$. This yields the differential equation for $G(x, \bar{x})$:

$$\begin{aligned} \big[ 64g^2(x-1)^2 x^2 \partial_x^2 + 16g(x-1)x(-2g^2(7x+4) + g(23x+2) - 6x)\partial_x \\ + (3g-2)\big(16(g-1)g^2 + 12(1-2g)gx + 3(5g-6)(1-2g)^2 x^2\big) \big] G(x, \bar{x}) = 0. \end{aligned} \tag{123}$$

Its Riemann scheme is:

$$\begin{array}{ccc} 0 & 1 & \infty \\ \hline \dfrac{2-3g}{2} & \dfrac{6g^2 - 13g + 6}{8g} & \dfrac{-6 + 17g - 10g^2}{8g} \\[2ex] \dfrac{1-g}{2} & \dfrac{38g^2 - 29g + 6}{8g} & \dfrac{-18g^2 + 21g - 6}{8g} \end{array} \tag{124}$$

These exponents correspond to the fusion rules

$$\tau_{\mathbb{1}} \times \tau_{\mathbb{1}} \to \mathbb{1} + \phi_{31} \otimes \phi_{31} + \ldots \qquad \text{and} \qquad \tau_{\mathbb{1}} \times \Phi \to \tau_{\mathbb{1}} + \tau_{\phi_{31}} + \ldots \tag{125}$$

in the channels $x \to 1$ and $x \to 0$, respectively.

**Solution.** The problem can be solved as in Sec. B.1. If we multiply the correlation function by the appropriate factor, equation (123) becomes the hypergeometric differential equation:

$$x(x-1)\partial_x^2 f + [(a+b+1)x - c]\partial_x f + ab f = 0,$$

$$\text{where}: f(x, \bar{x}) = |1-x|^{4\widehat{h}_{\mathbb{1}}} |x|^{4h_{21}} G(x, \bar{x}),$$

$$\text{and}: a = 2 - 3g, \qquad b = \frac{3}{2} - 2g, \qquad c = \frac{3}{2} - g. \tag{126}$$

Following exactly the reasoning in Sec. B.2, we find the correlation function:

$$G(x, \bar{x}) = |x|^{-4h_{21}} \Big[ \left| (1-x)^{-4\widehat{h}_{\mathbb{1}}} {}_2F_1(a, b; 1-d | 1-x) \right|^2 \tag{127}$$

$$+ X \left| (1-x)^{-4\widehat{h}_{\mathbb{1}} + 4g - 2} {}_2F_1(c-b, c-a; 1+d | 1-x) \right|^2 \Big], \tag{128}$$

where $d = c - a - b$ and

$$X = \frac{2^5(1-2g)^2\gamma\left(2g-\frac{1}{2}\right)^3\gamma(2-4g)^2}{(1-4g)^3\gamma(2-3g)}. \tag{129}$$

Of course, we may compare this solution to the one obtained by a conformal mapping of the four-point function $\langle\phi_{21}\phi_{21}\phi_{21}\phi_{21}\rangle$. The latter also satisfies a hypergeometric equation, and using the transformation 166, the two solutions can be shown to match.

## 5.5 Excited state entropy for minimal models at $N = 3$

We now consider the correlation function in the $N = 3$ orbifold of the minimal model $\mathcal{M}(p, p')$:

$$G(x, \bar{x}) = \langle\Phi(\infty)\tau_{\mathbb{1}}(1)\tau_{\mathbb{1}}(x, \bar{x})\Phi(0)\rangle, \tag{130}$$

where $\Phi = \phi_{21} \otimes \phi_{21} \otimes \phi_{21}$. The conformal mapping method would result in a much more complicated 6-point function, which is not the solution of an ordinary differential equation. But through the orbifold Virasoro structure, we can obtain such an equation. The method is similar in spirit to what was done for $N = 2$, finding the null vector conditions on the field $\Phi$, then using the Ward identities.

**Null vectors.** We will need the null vectors of $\Phi_3$ up to level 4:

Level 2: $\quad \widehat{L}^{(1)}_{-1}\widehat{L}^{(2)}_{-1}\Phi_3 = \frac{1}{2}\left(3g\widehat{L}^{(0)}_{-2} - \widehat{L}^{(0)}_{-1}\widehat{L}^{(0)}_{-1}\right)\Phi_3$

Level 3: $\quad \left[\widehat{L}^{(2)}_{-1}\widehat{L}^{(2)}_{-1}\widehat{L}^{(2)}_{-1} - 3g\left(\widehat{L}^{(1)}_{-2}\widehat{L}^{(2)}_{-1} - \widehat{L}^{(0)}_{-1}\widehat{L}^{(0)}_{-2}\right) + \widehat{L}^{(0)}_{-1}\widehat{L}^{(0)}_{-1}\widehat{L}^{(0)}_{-1}\right]\Phi_3 = 0$

Level 4: $\quad \left[3L^{(0)}_{-4} + L^{(0)}_{-3}L^{(0)}_{-1} + 3L^{(1)}_{-3}L^{(2)}_{-1} - L^{(2)}_{-3}L^{(1)}_{-1} - \frac{3g}{2}\left(L^{(0)}_{-2}\right)^2 + L^{(0)}_{-2}\left(L^{(0)}_{-1}\right)^2\right.$

$$\left. + 2L^{(2)}_{-2}\left(L^{(2)}_{-1}\right)^2 - \frac{1}{6g}\left(L^{(0)}_{-1}\right)^4 - \frac{1}{3g}L^{(0)}_{-1}\left(L^{(2)}_{-1}\right)^3\right]\Phi_3 = 0 \tag{131}$$

**Ward identities.** The descendants involving $\widehat{L}^{(r)}_{-1}$, with $r \neq 0$ are eliminated through the Ward identities. For example, using 77, with indices $(m_1, m_2, m_3, m_4) = (0, 1/3, -1/3, -2)$, and the function $\mathcal{G}(x, \bar{x}, z) = \langle\Phi_3|\widetilde{\tau}(1)\tau(x, \bar{x})T^{(2)}(z)\widehat{L}^{(2)}_{-1}\widehat{L}^{(2)}_{-1}|\Phi_3\rangle$, we obtain the relation:

$$\sum_{m=-2}^{2} Q_m(x)\langle\Phi|\widetilde{\tau}(1)\tau(x, \bar{x})\widehat{L}^{(2)}_m\widehat{L}^{(2)}_{-1}\widehat{L}^{(2)}_{-1}|\Phi\rangle = 0,$$

$$Q_{-2}(x) = 3^5 x^4, \qquad Q_{-1}(x) = 2 \cdot 3^4 x^3(2 + x), \qquad Q_0(x) = 27x^2(x^2 - 8x - 2),$$

$$Q_1(x) = 12x(x - 1)^3, \qquad Q_2(x) = (x - 1)^3(5 + 7x). \tag{132}$$

A similar relation can be found for the correlation functions of the form

$$\langle\Phi|\widetilde{\tau}(1)\tau(x, \bar{x})\widehat{L}^{(1)}_{-m}\widehat{L}^{(2)}_{-1}|\Phi\rangle \quad \text{and} \quad \langle\Phi|\widetilde{\tau}(1)\tau(x, \bar{x})\widehat{L}^{(2)}_{-m}\widehat{L}^{(1)}_{-1}|\Phi\rangle,$$

by using other Ward identities.

**Differential equation.** Putting everything together we find the following differential equation:

$$\left[P_0(\theta) + x P_1(\theta) + x^2 P_2(\theta) + x^3 P_3(\theta) + x^4 P_4(\theta)\right]G(x, \bar{x}) = 0, \tag{133}$$

where $\theta = x\partial_x$ and:

$$P_0(\theta) = 16(1-2g+g\theta)(1-g+g\theta)(6-5g+3g\theta)(6-4g+3g\theta),$$
$$P_1(\theta) = -16(1-g+g\theta)\big(486-963g+666g^2-160g^3$$
$$+(567g-810g^2+296g^3)\theta+(234g^2-180g^3)\theta^2+36g^3\theta^3\big),$$
$$P_2(\theta) = 28980-68076g+60344g^2-24320g^3+3840g^4$$
$$+(46008g-84456g^2+52272g^3-10944g^4)\theta \tag{134}$$
$$+(27720g^2-35280g^3+11424g^4)\theta^2+(7776g^3-5184g^4)\theta^3+864g^4\theta^4,$$
$$P_3(\theta) = -4(7-4g+2g\theta)\big(1215-1962g+1008g^2-160g^3$$
$$+(1296g-1404g^2+376g^3)\theta+(504g^2-288g^3)\theta^2+72g^3\theta^3\big),$$
$$P_4(\theta) = (7-4g+2g\theta)(7-2g+2g\theta)(15+-10g+6g\theta)(15-8g+6g\theta).$$

The Riemann scheme is given by:

$$
\begin{array}{ccc}
0 & 1 & \infty \\
\hline
\dfrac{-1+g}{g} & \dfrac{3}{2g} & \dfrac{15-8g}{6g} \\[2mm]
\dfrac{-6+4g}{3g} & \dfrac{-9+6g}{2g} & \dfrac{7-4g}{2g} \\[2mm]
\dfrac{-1+2g}{g} & \dfrac{-1+2g}{2g} & \dfrac{7-2g}{2g} \\[2mm]
\dfrac{-6+5g}{3g} & \dfrac{-5+4g}{2g} & \dfrac{15-10g}{6g}
\end{array}
\tag{135}
$$

**Interpretation in terms of OPEs.** The conformal dimensions of the internal field in the channels $x \to 1$ and $x \to 0$ are respectively:

$$\{0, h_{31}, 2h_{31}, 3h_{31}\} \qquad \text{and} \qquad \{\widehat{h}_{\phi_{21}}, \widehat{h}_{\phi_{21}} + 1/3, \widehat{h}_{\phi_{41}}, \widehat{h}_{\phi_{41}} + 1\}. \tag{136}$$

Since $\langle \tau_{\mathbb{1}}.\widetilde{\tau}_{\mathbb{1}}.(\phi_{31} \otimes \mathbb{1} \otimes \mathbb{1})\rangle \propto \langle \phi_{31}\rangle_{\mathbb{C}} = 0$, the conformal block with internal dimension $h_{31}$ is in fact not present in the physical correlation function. In the channel $x \to 0$, this is mirrored by the presence of two fields separated by an integer value : there is a degeneracy for the field $\tau_{\phi_{41}}$.

**This can be verified by applying the bootstrap method.** The equation has 4 linearly independent solutions, which can be computed by series expansion around the three singularities. Like in 5.3, the monodromy problem can be solved by comparing the expansions in their domain of convergence. Up to machine precision the coefficient corresponding to $h_{\phi_{3,1}}$ vanishes for all central charges. Nevertheless, the other structure constants converge, and we can still compute the full correlation function through bootstrap. For the non-zero structure constants, we checked that they were matching their theoretical expressions for simple minimal models (Yang-Lee and Ising).

The effective presence of only three conformal blocks also seem to imply that we should have been able to find a degree three differential equation, instead of four, for this correlation function. However we have not managed to derive such a differential equation.

# 6 Twist operators in critical RSOS models

In this section we describe a lattice implementation of the twist fields in the lattice discretisation of the minimal models, namely the critical Restricted Solid-On-Solid (RSOS) models. Entanglement entropy in RSOS models has already been considered in [47] for unitary models and in [39] for non-unitary models, but for a semi-infinite interval and away from criticality.

## 6.1 The critical RSOS model

Let us define the critical RSOS model with parameters $(m, k)$, where $m$ and $k$ are coprime integers and $k < m$. Each site $\vec{r}$ of the square lattice carries a height variable $a_{\vec{r}} \in \{1, 2, \ldots, m\}$, and two variables $a$ and $b$ sitting on neighbouring sites should differ by one : $|a - b| = 1$. The Boltzmann weight of a height configuration is given by the product of face weights:

$$W\left(\begin{array}{cc} a & b \\ d & c \end{array} \middle| u\right) = \begin{array}{c} a \quad\quad b \\ \boxed{u} \\ d \quad\quad c \end{array} = \sin(\lambda - u)\,\delta_{bd} + \sin u\,\delta_{ac}\,\frac{\sin \lambda\, b}{\sin \lambda\, a}\,, \tag{137}$$

where the crossing parameter $\lambda$ is

$$\lambda = \frac{\pi k}{m+1}\,. \tag{138}$$

The quantum model associated to the critical RSOS model is obtained by taking the very anisotropic limit $u \to 0$ of the transfer matrix. For periodic boundary conditions, one obtains a spin chain with basis states $|a_1, a_2, \ldots, a_L\rangle$, where $a_i \in \{1, \ldots, m\}$, and $|a_i - a_{i+1}| = 1$, and the Hamiltonian is

$$H_{\text{RSOS}} = -\sum_{i=1}^{L} e_i\,, \tag{139}$$

where $e_i$ only acts non-trivially on the heights $a_{i-1}, a_i, a_{i+1}$:

$$e_i|\ldots, a_{i-1}, a_i, a_{i+1}, \ldots\rangle = \delta_{a_{i-1}, a_{i+1}} \sum_{a_i', |a_i' - a_{i-1}| = 1} \frac{\sin \lambda\, a_i'}{\sin \lambda\, a_i} |\ldots, a_{i-1}, a_i', a_{i+1}, \ldots\rangle\,, \tag{140}$$

and the indices $i \pm 1$ are considered modulo $L$.

For simplicity, we now consider the RSOS model on a planar domain. The lattice partition function $Z_{\text{RSOS}}$ and the correlation functions admit a graphical expansion [48] in terms of non-intersecting, space-filling, closed loops on the dual lattice. The expansion of $Z_{\text{RSOS}}$ is obtained by associating a loop plaquette to each term in the face weight (137) as follows:

$$W\left(\begin{array}{cc} a & b \\ d & c \end{array} \middle| u\right) = \sin(\lambda - u)\, \begin{array}{c} a \quad b \\ d \quad c \end{array} + \sin u\, \begin{array}{c} a \quad b \\ d \quad c \end{array}\,. \tag{141}$$

Then, after summing on the height variables, each closed loop gets a weight $\beta = 2\cos\lambda$. Furthermore, following [49], correlation functions of the local variables

$$\varphi_q(a) = \frac{\sin \frac{\pi q a}{m+1}}{\sin \lambda\, a}\,, \qquad q \in \{1, \ldots, m\}\,, \tag{142}$$

also fit well in this graphical expansion. Let us recall, for example, the expansion of the one-point function $\langle \varphi_q(a_{\vec{r}}) \rangle$. For any loop which does not enclose $\vec{r}$, the height-dependent factors

from (137) end up to $\sin \lambda b / \sin \lambda a$, where $a$ (resp. $b$) is the outer (resp. inner) height adjacent to the loop. Thus, summing on the inner height $b$ gives the loop weight:

$$\sum_b A_{ab} \times \frac{\sin \lambda b}{\sin \lambda a} = 2 \cos \lambda = \beta \,, \tag{143}$$

where we have introduced the adjacency matrix $A_{ab} = 1$ if $|a - b| = 1$, and $A_{ab} = 0$ otherwise. For the loop enclosing $\vec{r}$ and adjacent to it, the factor $\varphi_q(b)$ should be inserted into the above sum, which gives:

$$\sum_b A_{ab} \times \varphi_q(b) \times \frac{\sin \lambda b}{\sin \lambda a} = \beta_q \times \varphi_q(a) \,, \tag{144}$$

where

$$\beta_q = 2 \cos\left(\frac{\pi q}{m+1}\right) . \tag{145}$$

Repeating this argument recursively, in the graphical expansion of $\langle \varphi_q(a_{\vec{r}}) \rangle$, one gets a loop weight $\beta_q$ for each loop enclosing the point $\vec{r}$. The $N$-point functions of the $\varphi_q$'s are treated similarly, through the use of a lattice Operator Product Expansion (OPE) [49].

This critical RSOS model provides a discretisation of the minimal model $\mathcal{M}(p, p')$, with central charge and conformal dimensions:

$$c = 1 - \frac{6(p - p')^2}{pp'} \,, \tag{146}$$

$$h_{rs} = \frac{(pr - p's)^2 - (p - p')^2}{4pp'} \,, \qquad r \in \{1, \ldots, p' - 1\}, \quad s \in \{1, \ldots, p - 1\}, \tag{147}$$

with the identification of parameters:

$$p = m + 1, \qquad p' = m + 1 - k \,. \tag{148}$$

The operator $\varphi_q$ changes the loop weight to $\beta_q$: thus, in this sector in the scaling limit, the dominant primary operator, which we denote $\phi_q$, has conformal dimension

$$h_{\phi_q} = \frac{q^2 - (p - p')^2}{4pp'} \,. \tag{149}$$

It is then easy to show[5] that $h_{\phi_q}$ is one of the dimensions of the Kac table (147). Note that $h_{\phi_{q=k}} = 0$ corresponds to the identity operator.

## 6.2 Partition function in the presence of branch points

We consider the reduced density matrix $\rho_{\mathscr{A}}$ for the interval $\mathscr{A} = [i, j]$. A generic matrix element $\langle a_i, a_{i+1}, \ldots, a_j | \rho_{\mathscr{A}} | a_i', a_{i+1}', \ldots, a_j' \rangle$ corresponds to the partition function of the lattice shown in Fig. 1, with the heights $a_i, \ldots a_j$ and $a_i', \ldots a_j'$ fixed, and the other heights summed over. In this convention, a branch point (or twist operator) sits on a site $\vec{r}$ of the square lattice, and is denoted $t(\vec{r})$. Computing the $n$-th Rényi entropy (2) amounts to determining the partition function $Z_{\text{RSOS}}^{(n)}$ on the surface obtained by "sewing" cyclically $n$ copies of the diagram in Fig. 1 along the cut going from $a_i$ to $a_j$.

The graphical expansion of the partition function on this surface with two branch points is very similar to the case of a planar domain. The only difference concerns the loops which

---

[5]Since $p$ and $p'$ are coprime, using the Bézout theorem, there exist two integers $u$ and $v$ such that $pu - p'v = 1$, and then it is possible to find an integer $\ell$ such that $(r, s) = (qu + \ell p', qv + \ell p)$ belongs to the range (147), whereas $pr - p's = q$.

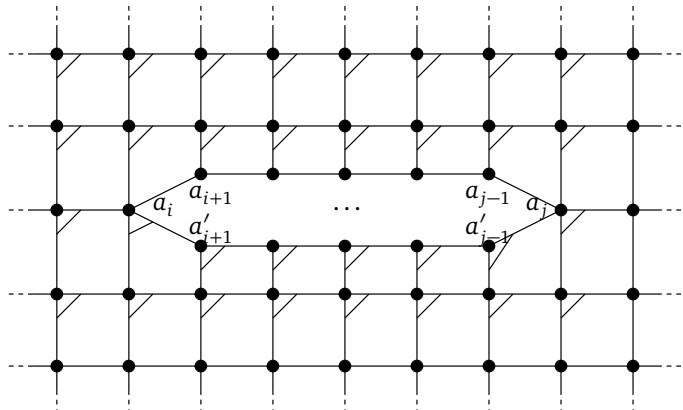

Figure 1: A generic reduced density matrix element $\langle a_i, a_{i+1}, \ldots, a_j | \rho_{\mathscr{A}} | a'_i, a'_{i+1}, \ldots, a'_j \rangle$ for the interval $\mathscr{A} = [i, j]$. This matrix element is set to zero if $a_i \neq a'_i$ or $a_j \neq a'_j$.

surround one branch point. Since such a loop has a total winding $\pm 2\pi n$ instead of $\pm 2\pi$, the height-dependent factors from (137) end up to $(\sin \lambda b / \sin \lambda a)^n$, where $a$ (resp. $b$) is the external (resp. internal) height adjacent to the loop. Since $(\sin \lambda b)^n$ is not an eigenvector of the adjacency matrix $A$, the sum over $b$ does not give a well-defined loop weight.

For this reason, we introduce a family of modified lattice twist operators:

$$t_q(\vec{r}) = \widehat{\varphi}_q(a_{\vec{r}}) \times t(\vec{r}), \qquad \text{where} \quad \widehat{\varphi}_q(b) = \frac{\sin \frac{\pi q b}{m+1}}{(\sin \lambda b)^n}. \tag{150}$$

With this insertion of $\widehat{\varphi}_q$ at the position of the twist, the sum over the internal height gives:

$$\sum_b A_{ab} \left( \frac{\sin \lambda b}{\sin \lambda a} \right)^n \times \widehat{\varphi}_q(b) = \beta_q \times \widehat{\varphi}_q(a), \tag{151}$$

and hence any loop surrounding $t_q$ gets a weight $\beta_q$ (145). Thus, the scaling limit of $t_q$ is the primary operator $\tau_{\phi_q}$, which belongs to the twisted sector, and has conformal dimension

$$\widehat{h}_{\phi_q} = \frac{c}{24} \left( n - \frac{1}{n} \right) + \frac{h_{\phi_q}}{n}. \tag{152}$$

In particular, since $\beta_k = \beta = 2 \cos \lambda$, one has $\phi_k = \mathbb{1}$, and the lattice operator $t_k$ corresponds to $\tau_{\mathbb{1}}$ in the scaling limit. Note that the "bare" twist operator $t$ is itself a linar combination of the $t_q$'s:

$$t(\vec{r}) = \sum_{q=1}^m x_q \, t_q(\vec{r}), \qquad \text{where} \quad x_q = \frac{2}{m} \sum_{a=1}^m (\sin \lambda a)^n \sin \left( \frac{\pi q a}{m+1} \right). \tag{153}$$

The scaling limit of $t$ is thus always determined by the term $t_1$, since it has the lowest conformal dimension. In the case of a unitary minimal model ($k = 1$), this corresponds to $\tau_{\mathbb{1}}$. In contrast, for a non-unitary minimal model $\mathscr{M}(p, p')$, $t$ scales to the twist operator $\tau_{\phi_1}$, where $\phi_1$ is the primary operator with the lowest (negative) conformal dimension in the Kac table: $h_{\phi_1} = -[(p - p')^2 - 1]/(4pp')$.

## 6.3 Rényi entropies of the RSOS model

When defining a zero-temperature Rényi entropy, two distinct parameters must be specified:

1. The state $|\psi\rangle_R$ in which the entropy is measured (or equivalently the density matrix $\rho = |\psi\rangle_R\langle\psi|_L$).

2. The local state $\phi_q$ of the system in the vicinity of branch points. This determines which twist operator $t_q$ should be inserted. In the case of the physical Rényi entropy defined as (2), one inserts the linear combination $t(\vec{r}) = \sum_{q=1}^m x_q t_q(\vec{r})$.

In the following, we will be interested in the Rényi entropy of an interval of length $\ell$ in the spin chain $H_{\text{RSOS}}$ (139) of length $L$ with periodic boundary conditions. This corresponds to the lattice average value:

$$\frac{1}{1-N} \log\langle\Psi|t_q(0)t_q(\ell)|\Psi\rangle , \tag{154}$$

where $|\Psi\rangle = |\psi\rangle^{\otimes N}$. The "physical" Rényi entropy (2) is related to:

$$\frac{1}{1-N} \log\langle\Psi|t(0)t(\ell)|\Psi\rangle . \tag{155}$$

The average values (154–155) scale to correlation functions on the cylinder $\{z\,|\,0 \le \text{Im}\,z < L\}$:

$$\langle\Psi|t_q(0)t_q(\vec{u})|\Phi\rangle \propto \langle\Phi(-\infty)\tau_{\phi_q}(0)\tau_{\phi_q}(u,\bar{u})\Psi(+\infty)\rangle_{\text{cyl}} , \tag{156}$$

where $u = i\ell$, and similarly for $t_q$ replaced by $t$. Using the conformal map $x = \exp(2\pi u/L)$, these are related to the correlation functions on the complex plane:

$$\langle\Psi(-\infty)\tau_{\phi_q}(0)\tau_{\phi_q}(u,\bar{u})\Psi(+\infty)\rangle_{\text{cyl}} = \left(\frac{2\pi}{L}\right)^{4\widehat{h}_{\phi_q}} \langle\Psi(0)\tau_{\phi_q}(1)\tau_{\phi_q}(x,\bar{x})\Psi(\infty)\rangle_{\text{plane}} . \tag{157}$$

In the case of the (generalised) Rényi entropy in the vacuum, we have $\psi = \mathbb{1}$, and this becomes a two-point function, which is easily evaluated:

$$S_N(x,\mathbb{1},\tau_{\phi_q}) = \frac{4\widehat{h}_{\phi_q}}{N-1} \log\left(\frac{L}{\pi}\sin\frac{\pi\ell}{L}\right) + \text{const} . \tag{158}$$

In particular, when $\phi_q = \mathbb{1}$, one recovers the result from [14]:

$$S_N(x,\mathbb{1},\tau_{\mathbb{1}}) = \frac{(N+1)c}{6N} \log\left(\frac{L}{\pi}\sin\frac{\pi\ell}{L}\right) + \text{const} . \tag{159}$$

For a generic state $|\psi\rangle$ however, the entropy $S_N(x,\psi,\tau_{\phi_q})$ remains a non-trivial function of $\ell$, and does not reduce to the simple form (158).

## 6.4 Numerical computations

### 6.4.1 Numerical setup

We have computed some Rényi entropies (154) and (155) in the RSOS model with parameters $m = 4$ and $k = 3$, corresponding to the Yang-Lee singularity $\mathcal{M}(5,2)$ with central charge $c = -22/5$. The primary fields are $\mathbb{1} = \phi_{11} \equiv \phi_{14}$ and $\phi = \phi_{12} \equiv \phi_{13}$, with conformal dimensions $h_{\mathbb{1}} = 0$ and $h_{\phi} = -1/5$. They correspond respectively to $\mathbb{1} \propto \varphi_3$ and $\phi \propto \varphi_1$.

A lattice eigenvector (scaling either to $|\mathbb{1}\rangle$, or to $|\phi\rangle$) of $H_{\text{RSOS}}$ with periodic boundary conditions is obtained by exact diagonalisation with the QR or Arnoldi method, and then used to construct the reduced density matrix $\rho_{\mathscr{A}}$, where $\mathscr{A} = [0,\ell]$. For the computation of $S_N(x,\mathbb{1},t_q)$ and $S_N(x,\psi,t_q)$, the factor $\widehat{\varphi}_q(a_0)\widehat{\varphi}_q(a_\ell)$ [see (150)] is inserted into the trace (2) of $\rho_{\mathscr{A}}^N$. For the computation of $S_N(x,\phi,t)$, no additional factor is inserted. From the above discussion, we expect $S_N(x,\phi,t)$ to be described by the insertion of the dominant twist operators $\tau_{\phi}$.

### 6.4.2 Results for entropies at $N = 2$ in the Yang-Lee model

Here we present our numerical results obtained with the procedure described above. In all the cases considered the cylinder correlation functions have been rescaled with the factor $L^{4h}$, where $h$ is the appropriate twist field conformal dimension. The collapse of various finite size data further confirms the correct identification of the twist field ($\tau_\phi$ or $\tau_{\mathbb{1}}$).

The results obtained are in excellent agreement with our CFT interpretation (18) and with our analytical results. Moreover they are clearly not compatible (see Fig. 3) with the claim

$$S_N \sim \frac{c_{\text{eff}}}{6} \frac{N+1}{N} \log |u - v|, \tag{160}$$

which can be found in the literature [33, 41] (the effective central charge of the Yang-Lee model is $c_{\text{eff}} = 2/5$).

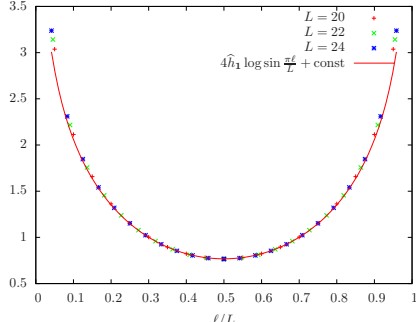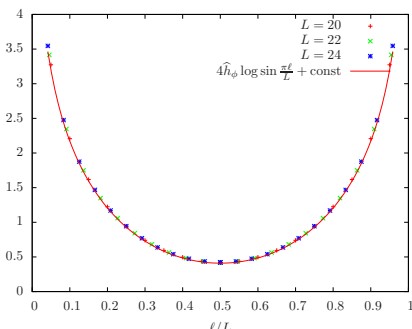

Figure 2: $N = 2$ Rényi entropy of the Yang-Lee model in the vacuum $|0\rangle$ with various twist fields. In the left panel we consider the twist $t_3$ as in eq. (154), which corresponds in the continuum to $\tau_{\mathbb{1}}$ (with $\widehat{h}_{\mathbb{1}} = -\frac{11}{40}$). In the right panel the bare twist $t$ is considered (eq. (155)), corresponding to $\tau_\phi$ (with $\widehat{h}_\phi = -\frac{3}{8}$).

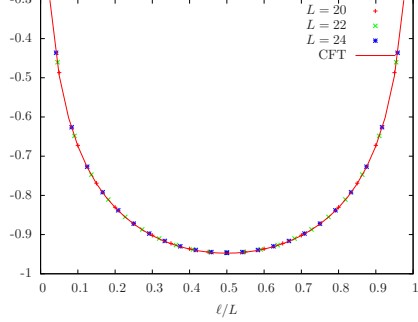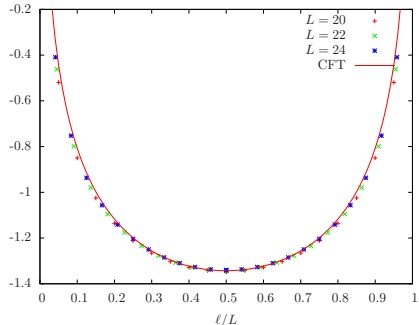

Figure 3: $N = 2$ Rényi entropy of the Yang-Lee model in the ground state $|\phi\rangle$ with various twist fields. In the left panel we consider the twist $t_3$ as in eq. (154), which corresponds in the continuum to $\tau_{\mathbb{1}}$. Exact diagonalisation results are compared to the CFT prediction (190) for the function $\langle \Phi(0) \tau_{\mathbb{1}}(1) \tau_{\mathbb{1}}(x, \bar{x}) \Phi(\infty) \rangle$. In the right panel the bare twist $t$ is considered (eq. (155)). Exact diagonalisation results are compared to the CFT prediction (92) for the function $\langle \Phi(0) \tau_\phi(1) \tau_\phi(x, \bar{x}) \Phi(\infty) \rangle$.

### 6.4.3 Results for entropies at $N = 3$ in the Yang-Lee model

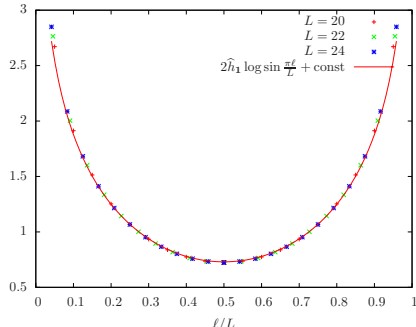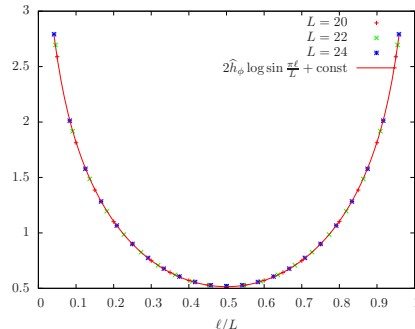

Figure 4: $N = 3$ Rényi entropy of the Yang-Lee model in the vacuum $|0\rangle$ with various twist fields. In the left panel we consider the twist $t_3$ as in eq. (154), which corresponds in the continuum to $\tau_{\mathbb{1}}$ (with $\widehat{h}_{\mathbb{1}} = -\frac{22}{45}$). In the right panel the bare twist $t$ is considered (eq. (155)), corresponding to $\tau_\phi$ (with $\widehat{h}_\phi = -\frac{5}{9}$).

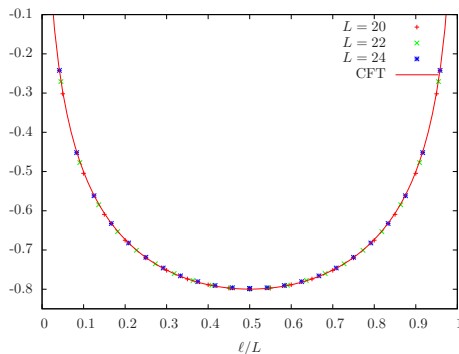

Figure 5: The $N = 3$ Rényi entropy (154) of the Yang-Lee model in the state $|\phi\rangle$, with the twist $t_3$ corresponding to $\tau_{\mathbb{1}}$. Exact diagonalisation results are compared to the CFT prediction from Sec. 5.5 for the function $\langle \Phi(0)\tau_{\mathbb{1}}(1)\tau_{\mathbb{1}}(x, \bar{x})\Phi(\infty)\rangle$.

## 7 Conclusion

In this article we have studied the Rényi entropies of one-dimensional critical systems, using the mapping of the $N^{\text{th}}$ Rényi entropy to a correlation function involving twist fields in a $\mathbb{Z}_N$ cyclic orbifold. When the CFT describing the universality class of the critical system is rational, so is the corresponding cyclic orbifold. It follows that the twist fields are degenerate : they have null vectors. From these null vectors a Fuchsian differential equation is derived, although this step can be rather involved since the null-vector conditions generically involve fractional modes of the orbifold algebra. The last step is to solve this differential equation and build a monodromy invariant correlation function, which is done using standard bootstrap methods. We have exemplified this method with the calculation of various one-interval Rényi entropies in the Yang-Lee model, a two-interval entropy in the Ising model and some one-interval entropies computed in specific excited states for all minimal models.

We have also described a lattice implementation of the twist fields in the lattice discretisation of the minimal models, namely the critical Restricted Solid-On-Solid (RSOS) models.

This allows us to check numerically our analytical results obtained in the Yang-Lee model. Excellent agreement is found.

The main limitation of our method is that its gets more involved as $N$ increases, and as the minimal model $\mathcal{M}(p, p')$ under consideration gets more complicated (*i.e.* as $p$ and/or $p'$ increases). For this reason, we have limited our study to $N = 2$ and $N = 3$ in the Yang-Lee model $\mathcal{M}(5, 2)$. However, this method is applicable in a variety of situations where no other method is available, for instance when the subsystem $\mathcal{A}$ is not connected (*e.g.* two-intervals EE). We will address this case in a following paper.

Another interesting research direction would be to develop a Coulomb Gas formalism for the cyclic orbifold, as it would provide an efficient tool to solve the twist-field differential equations *à la* Dotsenko-Fateev. Indeed, the Coulomb Gas yields a very natural way to write down conformal blocks (in the form of closed contour integrals of screening operators), to compute their monodromies, and from there to solve the bootstrap.

## Acknowledgements

The authors wish to thank Olivier Blondeau-Fournier, Jérôme Dubail, Benjamin Doyon and Olalla Castro-Alvaredo for valuable discussions.

## Appendix

## A   Properties of hypergeometric functions

- The Gauss hypergeometric function $_2\mathrm{F}_1$ is defined as:

$$_2\mathrm{F}_1(a, b; c | x) = \sum_{n=0}^{\infty} \frac{(a)_n (b)_n}{n! (c)_n} x^n, \tag{161}$$

  where $(a)_n = a(a + 1) \ldots (a + n - 1)$ is the Pochhammer symbol.

- Under the transformation $x \mapsto 1 - x$ of the complex variable, it satisfies the relations:

$$\frac{1}{\Gamma(c)} {}_2\mathrm{F}_1(a, b; c | x) = \frac{\Gamma(d)}{\Gamma(c - a)\Gamma(c - b)} {}_2\mathrm{F}_1(a, b, 1 - d | 1 - x)$$
$$+ \frac{\Gamma(-d)}{\Gamma(a)\Gamma(b)} (1 - x)^d {}_2\mathrm{F}_1(c - a, c - b; 1 + d | 1 - x), \tag{162}$$
$${}_2\mathrm{F}_1(a, b; 1 - d | 1 - x) = x^{1-c} {}_2\mathrm{F}_1(a - c + 1, b - c + 1; 1 - d | 1 - x), \tag{163}$$

  for $d = c - a - b$.

- The matrix relating the bases of solutions to the hypergeometric differential equation $\{I_1, I_2\}$ (57) and $\{J_1, J_2\}$ (178) as

$$I_i(x) = \sum_{j=1}^{2} A_{ij} J_j(x) \tag{164}$$

is given by:

$$
A = \begin{bmatrix} \frac{\Gamma(c)\Gamma(d)}{\Gamma(c-a)\Gamma(c-b)} & \frac{\Gamma(c)\Gamma(-d)}{\Gamma(a)\Gamma(b)} \\ \frac{\Gamma(2-c)\Gamma(d)}{\Gamma(1-a)\Gamma(1-b)} & \frac{\Gamma(2-c)\Gamma(-d)}{\Gamma(1-c+a)\Gamma(1-c+b)} \end{bmatrix},
$$
$$
A^{-1} = \begin{bmatrix} \frac{\Gamma(1-c)\Gamma(1-d)}{\Gamma(1-c+a)\Gamma(1-c+b)} & \frac{\Gamma(c-1)\Gamma(1-d)}{\Gamma(a)\Gamma(b)} \\ \frac{\Gamma(1-c)\Gamma(1+d)}{\Gamma(1-a)\Gamma(1-b)} & \frac{\Gamma(c-1)\Gamma(1+d)}{\Gamma(c-a)\Gamma(c-b)} \end{bmatrix}.
\tag{165}
$$

- Under the transformation $x \mapsto 4\sqrt{x}/(1+\sqrt{x})^2$, we have:

$$
{}_2F_1(a, b; a-b+1 | x) =
$$
$$
(1+\sqrt{x})^{-2a} \, {}_2F_1\left(a, a-b+\frac{1}{2}; 2a-2b+1 \,\bigg|\, \frac{4\sqrt{x}}{(1+\sqrt{x})^2}\right).
\tag{166}
$$

# B Four-point function satisfying a second-order differential equation

In this appendix we compute the correlation function

$$
G(x, \bar{x}) = \langle \tau_{\mathbb{1}}(\infty) \tau_{\mathbb{1}}(1) \tau_{\mathbb{1}}(x, \bar{x}) \tau_{\mathbb{1}}(0) \rangle.
\tag{167}
$$

in the $\mathbb{Z}_2$ orbifold of the YL model. It follows from the null-vector

$$
\left(\widehat{L}_{-2} - \frac{10}{3}\widehat{L}_{-1}^2\right)\tau_{\mathbb{1}} = 0.
\tag{168}
$$

This is the standard form of a null-vector at level 2, which yields in the usual way a second order differential equation whose solutions are hypergeometric functions. For completeness we recall the key steps in computing $G(x, \bar{x})$.

## B.1 Differential equation

A standard CFT argument yields, for any $n \in \mathbb{Z}$, any primary operators $(\mathcal{O}_2, \mathcal{O}_3)$, and any states $(|\mathcal{O}_1\rangle, |\mathcal{O}_4\rangle)$:

$$
\langle \mathcal{O}_1 | \mathcal{O}_2(1)\mathcal{O}_3(x, \bar{x})L_n | \mathcal{O}_4 \rangle - \langle \mathcal{O}_1 | L_n \mathcal{O}_2(1)\mathcal{O}_3(x, \bar{x}) | \mathcal{O}_4 \rangle
$$
$$
= \left\{ (1-x^n)[x\partial_x + (n+1)h_3] + (h_4 - h_1) - n(h_2 + h_3) \right\} \langle \mathcal{O}_1 | \mathcal{O}_2(1)\mathcal{O}_3(x, \bar{x}) | \mathcal{O}_4 \rangle.
\tag{169}
$$

Then (56) translates into the ordinary differential equation for $G_{(}x, \bar{x})$:

$$
\left[ 400(x-1)^2 x^2 \partial_x^2 + 40(x-1)x(6x-3)\partial_x + 33 \right] G(x) = 0.
\tag{170}
$$

This equation has the following Riemann scheme, giving the local exponents, i.e. the allowed power-law behaviours at the three singular points 0, 1 and $\infty$:

$$
\begin{array}{ccc}
0 & 1 & \infty \\
\hline
\frac{3}{20} & \frac{3}{20} & 0 \\
\frac{11}{20} & \frac{11}{20} & -\frac{2}{5}
\end{array}
\tag{171}
$$

In the limits $x \to 0, 1, \infty$, we have the OPE:

$$
\tau_{\mathbb{1}} \times \tau_{\mathbb{1}} \to \mathbb{1} + \Phi.
\tag{172}
$$

So the local exponents (171) are consistent with the internal states $\{\mathbb{1}, \Phi\}$ of the conformal blocks for the channels. If we perform the appropriate change of function to shift two of these local exponents to zero:

$$G(x, \bar{x}) = |x|^{11/10} |1-x|^{11/10} f(x, \bar{x}),\tag{173}$$

then (170) turns into the hypergeometric differential equation:

$$x(x-1)\partial_x^2 f + [(a+b+1)x - c]\partial_x f + ab\, f = 0,\tag{174}$$

with parameters:

$$a = \frac{7}{10}, \qquad b = \frac{11}{10}, \qquad c = \frac{7}{5}.\tag{175}$$

It is also convenient to introduce the parameter $d = c - a - b$. If one repeats the argument with the anti-holomorphic generators $\bar{L}_n$, one obtains the same equation as (174), with $(x, \partial_x)$ replaced by $(\bar{x}, \bar{\partial}_x)$.

## B.2 Determination of the four-point function

A basis of holomorphic solutions to (174) is given by:

$$\begin{aligned} I_1(x) &= {}_2F_1(a, b; c | x), \\ I_2(x) &= x^{1-c} \,{}_2F_1(b-c+1, a-c+1; 2-c | x), \end{aligned}\tag{176}$$

where ${}_2F_1$ is Gauss's hypergeometric function (161). The basis $I_j$ has a diagonal monodromy around $x = 0$:

$$\begin{pmatrix} I_1(x) \\ I_2(x) \end{pmatrix} \underset{x \mapsto e^{2i\pi}x}{\mapsto} \begin{pmatrix} 1 & 0 \\ 0 & e^{-2i\pi c} \end{pmatrix} \begin{pmatrix} I_1(x) \\ I_2(x) \end{pmatrix}.\tag{177}$$

Similarly, by the change of variable $x \mapsto 1-x$, one obtains a basis of solutions

$$\begin{aligned} J_1(x) &= {}_2F_1(a, b; a+b-c+1 | 1-x), \\ J_2(x) &= (1-x)^{c-a-b} \,{}_2F_1(c-b, c-a; c-a-b+1 | 1-x), \end{aligned}\tag{178}$$

with a diagonal monodromy around $x = 1$:

$$\begin{pmatrix} J_1(x) \\ J_2(x) \end{pmatrix} \underset{(1-x) \mapsto e^{2i\pi}(1-x)}{\mapsto} \begin{pmatrix} 1 & 0 \\ 0 & e^{2i\pi(c-a-b)} \end{pmatrix} \begin{pmatrix} J_1(x) \\ J_2(x) \end{pmatrix}.\tag{179}$$

We shall construct a solution of the form

$$G(x, \bar{x}) = |x|^{11/10} |1-x|^{11/10} \sum_{i,j=1}^{2} X_{ij} \overline{I_i(x)} I_j(x).\tag{180}$$

From the properties of the operators $\tau_{\mathbb{1}}$ and $\Phi$ (see Sec. 2), $G(x, \bar{x})$ should be single-valued, which imposes the form $X_{ij} = \delta_{ij} X_i$ for the coefficients in (180). The solution should also admit a decomposition of the form:

$$G(x, \bar{x}) = |x|^{11/10} |1-x|^{11/10} \sum_{k,\ell=1}^{2} Y_{k\ell} \overline{J_k(x)} J_\ell(x).\tag{181}$$

Again, single-valuedness of $G(x, \bar{x})$ imposes the form $Y_{k\ell} = \delta_{k\ell} Y_k$. The key ingredient to determine the coefficients $X_j$ and $Y_j$ is the matrix for the change of basis between $\{I_1(x), I_2(x)\}$

and $\{J_1(x), J_2(x)\}$. Using the properties (162–163) of hypergeometric functions, one obtains:

$$I_i(x) = \sum_{j=1}^{2} A_{ij} J_j(x), \tag{182}$$

where $A$ is given in (165). Comparing (180) and (181), we get the matrix relations:

$$Y = A^\dagger X A, \qquad X = (A^{-1})^\dagger Y A^{-1}. \tag{183}$$

Imposing a diagonal form for $X$ and $Y$ yields two linear equations on $(X_1, X_2)$:

$$\begin{aligned} \bar{A}_{11} X_1 A_{12} + \bar{A}_{21} X_2 A_{22} &= 0, \\ \bar{A}_{12} X_1 A_{11} + \bar{A}_{22} X_2 A_{21} &= 0. \end{aligned} \tag{184}$$

Since the entries of $A$ are real, these two relations are equivalent. Similarly, one gets a linear relation between $Y_1$ and $Y_2$. Finally, one gets the ratios:

$$\frac{X_2}{X_1} = -\left[\frac{\Gamma(c)}{\Gamma(2-c)}\right]^2 \gamma(1-a)\gamma(1-b)\gamma(1-c+a)\gamma(1-c+b), \tag{185}$$

$$\frac{Y_2}{Y_1} = -\left[\frac{\Gamma(1-d)}{\Gamma(1+d)}\right]^2 \gamma(1-a)\gamma(1-b)\gamma(c-a)\gamma(c-b). \tag{186}$$

The symbol $\Gamma$ denotes Euler's Gamma function, and we also introduced the short-hand notation:

$$\gamma(x) = \frac{\Gamma(x)}{\Gamma(1-x)}. \tag{187}$$

Moreover, the term $|J_1(x)|^2$ in (181) corresponds to the OPE $\tau_{\mathbb{1}} \times \tau_{\mathbb{1}} \to \mathbb{1}$, which fixes $Y_1 = 1$. We thus get:

$$X_1 = \gamma(1-c)\gamma(1-d)\gamma(c-a)\gamma(c-b) = 1, \quad X_2 = -\frac{\gamma(c)}{(1-c)^2}\gamma(1-d)\gamma(1-a)\gamma(1-b) = 2^{16/5}, \tag{188}$$

and

$$Y_1 = 1, \qquad Y_2 = -\left[\frac{\Gamma(1-d)}{\Gamma(1+d)}\right]^2 \gamma(1-a)\gamma(1-b)\gamma(c-a)\gamma(c-b) = 2^{16/5}. \tag{189}$$

The final result for the four-point function $G(x, \bar{x})$ (54) is

$$\begin{aligned} G(x, \bar{x}) = |x|^{11/10}\,|1-x|^{11/10} \times \Big[ &\left|{}_2F_1(7/10, 11/10; 7/5|x)\right|^2 \\ &+ 2^{16/5}\left|x^{-2/5}\,{}_2F_1(7/10, 3/10; 3/5|x)\right|^2 \Big], \end{aligned} \tag{190}$$

## C Direct computation of OPE coefficients of twist operators

In this appendix, we perform the computation of the structure constants appearing in the Yang-Lee model on the $N = 2$ orbifold. They provide a non-trivial check of the validity of our method based on solving the differential equation for conformal blocks. In the following, $\langle \ldots \rangle$ denotes the average in the orbifold theory, whereas $\langle \ldots \rangle_{\Sigma_2}$ (resp. $\langle \ldots \rangle_{\mathbb{C}}$) denotes the average in the mother theory on the two-sheeted Riemann surface (resp. on the Riemann sphere).

Some of those results were already obtained in [33], a generic way of computing those three-point functions can be found in [50]. In the specific case of the Ising model similar three-point functions were found in [19] and [32].

For three-point functions involving only untwisted operators, the correlation function decouples betwenn the $N$ copies. For instance:

$$C(\Phi, \Phi, \Phi) = C(\phi, \phi, \phi)^2, \tag{191}$$

where $C(\phi, \phi, \phi)$ is the structure constant in the mother theory (Yang-Lee):

$$C(\phi, \phi, \phi) = \frac{i\sqrt{\frac{1}{2}\left(3\sqrt{5} - 5\right)}\Gamma\left(\frac{1}{5}\right)^3}{10\pi\Gamma\left(\frac{3}{5}\right)}.$$

The structure constants involving twist operators can be computed by unfolding through the conformal map $z \mapsto z^{1/2}$.

- Let us start with $C(\Phi, \tau_{\mathbb{1}}, \tau_{\mathbb{1}})$, which unfolds to a two-point function:

$$\begin{aligned}
C(\Phi, \tau_{\mathbb{1}}, \tau_{\mathbb{1}}) &= \langle \tau_{\mathbb{1}}(\infty)\Phi(1)\tau_{\mathbb{1}}(0)\rangle \\
&= \langle \phi(1)\phi(e^{2i\pi})\rangle_{\Sigma_2} \\
&= \langle 2^{-2h_\phi}\phi(1) \times 2^{-2h_\phi}\phi(-1)\rangle_{\mathbb{C}} \\
&= 2^{-8h_\phi}.
\end{aligned} \tag{192}$$

- The constant $C[(\mathbb{1} \otimes \phi)^{(0)}, \tau_\phi, \tau_\phi]$ involving $(1 \otimes \phi)^{(0)} = \frac{1}{\sqrt{2}}(\mathbb{1} \otimes \phi + \phi \otimes \mathbb{1})$:

$$\begin{aligned}
C[(\mathbb{1} \otimes \phi)^{(0)}, \tau_\phi, \tau_\phi] &= \lim_{z_\infty \to \infty} z_\infty^{4h_\phi}\langle \tau_\phi(0)\tau_\phi(1)(\mathbb{1} \otimes \phi)^{(0)}(z_\infty)\rangle \\
&= \sqrt{2}\lim_{z_\infty \to \infty} z_\infty^{4h_\phi}\langle \tau_\phi(0)\tau_\phi(1)(\mathbb{1} \otimes \phi)(z_\infty)\rangle \\
&= \frac{\sqrt{2}}{2^{2h_\phi}}\lim_{z_\infty \to \infty} z_\infty^{4h_\phi}\langle \phi(0)\phi(1)\phi(z_\infty)\rangle_{\mathbb{C}} \\
&= \frac{\sqrt{2}C(\phi, \phi, \phi)}{2^{2h_\phi}} \approx 3.56664i.
\end{aligned} \tag{193}$$

- $C(\Phi, \tau_\phi, \tau_\phi)$ unfolds to a four-point function, computed in D:

$$\begin{aligned}
C(\Phi, \tau_\phi, \tau_\phi) &= \langle \tau_\phi | \tau_\phi(1) | \Phi\rangle \\
&= \frac{1}{2^{4h_\phi}}\langle \phi|\phi(1)\phi(-1)|\phi\rangle_{\mathbb{C}} \\
&= \frac{\left(\sqrt{5} - 1\right)\Gamma\left(\frac{1}{5}\right)^6\Gamma\left(\frac{2}{5}\right)^2}{80 \, 2^{2/5}\pi^4} \approx -5.53709.
\end{aligned} \tag{194}$$

- $C(\tau_\phi, \Phi, \tau_{\mathbb{1}})$:

$$\begin{aligned}
C(\tau_\phi, \Phi, \tau_{\mathbb{1}}) &= \langle \tau_\phi | \tau_{\mathbb{1}}(1) | \Phi\rangle \\
&= \frac{\langle \phi(0)\phi(1)\phi(-1)\rangle_{\mathbb{C}}}{2^{4h_\phi}} = \frac{C(\phi, \phi, \phi)}{2^{6h_\phi}} \approx 4.39104i.
\end{aligned} \tag{195}$$

- $C\left(\tau_\phi, \Phi, \widehat{L}_{-1/2}\widehat{\bar{L}}_{-1/2}\tau_\phi\right)$ : we also need this structure constant which involves a descendant state. The behaviour of the descendants states during the unfolding is given by the induction procedure [42]:

$$\widehat{L}_{-1/2} \to \frac{1}{2}L_{-1}.$$

Hence, for the three-point function:

$$\left\langle\left[\widehat{L}_{-1/2}\widehat{\bar{L}}_{-1/2}\tau_\phi\right](0)\tau_\phi(1)\Phi(\infty)\right\rangle = \frac{1}{2^{4h_\phi+2}}\langle\phi|\phi(1)\phi(-1)L_{-1}\bar{L}_{-1}|\phi\rangle.$$

This four-point function is computed in D. To compute the structure constant, we also need to normalize the descendant state:

$$
\begin{aligned}
C\left(\tau_\phi, \Phi, \widehat{L}_{-1/2}\widehat{\bar{L}}_{-1/2}\tau_\phi\right) &= \frac{\left\langle\left[\widehat{L}_{-1/2}\widehat{\bar{L}}_{-1/2}\tau_\phi\right](0)\tau_\phi(1)\Phi(\infty)\right\rangle}{\sqrt{\langle\tau_\phi|\widehat{L}_{1/2}\widehat{\bar{L}}_{1/2}\widehat{L}_{-1/2}\widehat{\bar{L}}_{-1/2}|\tau_\phi\rangle}} \\
&= \frac{10}{2^{4h_\phi+2}}\left[\partial_z\,\partial_{\bar{z}}\langle\phi(z)\phi(1)\phi(-1)\phi(\infty)\rangle|_{z=\bar{z}=0}\right] \\
&= \frac{2^{4h_\phi+2}}{5} \approx 0.459479.
\end{aligned}
$$
(196)

# D Direct computation of the function of Sec. 4

The correlation function $F(x,\bar{x})$ (54) can be computed using a direct approach, by relating it to the four-point function $\langle\phi(\infty)\phi(1)\phi(u)\phi(0)\rangle$ through an appropriate conformal mapping from the two-sheeted Riemann surface $\Sigma_2$ to the Riemann sphere. Indeed, let $y \in \mathbb{C}$, and consider the mapping:

$$z \mapsto w = \frac{2y(\sqrt{z}-1)}{(1+y)(\sqrt{z}-y)}, \qquad \frac{dw}{dz} = \frac{y(1-y)}{(1+y)(\sqrt{z}-y)^2\sqrt{z}}.$$
(197)

The function $F(x,\bar{x})$ can be written:

$$
\begin{aligned}
F(x,\bar{x}) &= \langle\tau(\infty)\Phi(1)\Phi(x,\bar{x})\tau(0)\rangle \\
&= \langle\tau(\infty)\tau(0)\rangle \times \langle\phi(1)\phi(e^{2i\pi})\phi(x,\bar{x})\phi(e^{2i\pi}x,e^{-2i\pi}\bar{x})\rangle_{\Sigma_2}.
\end{aligned}
$$
(198)

The four points of this correlation function are mapped as follows under (197):

$$1 \mapsto 0, \quad e^{2i\pi} \mapsto \frac{4y}{(1+y)^2}, \quad x \mapsto R = \frac{2y(1-\sqrt{x})}{(1+y)(y-\sqrt{x})}, \quad e^{2i\pi}x \mapsto \frac{2y(1+\sqrt{x})}{(1+y)(y+\sqrt{x})}.$$
(199)

If we let $y \to \sqrt{x}$, we have $R \to \infty$, and we get

$$F(x,\bar{x}) = |1+\sqrt{x}|^{-8h_\phi}|16x|^{-2h_\phi} \times \langle\phi(\infty)\phi(1)\phi(u,\bar{u})\phi(0)\rangle_{\mathbb{C}}, \qquad u = \frac{4\sqrt{x}}{(1+\sqrt{x})^2}.$$
(200)

Since $\phi \equiv \phi_{12}$ is degenerate at level 2 (see (46), the function $\langle\phi(\infty)\phi(1)\phi(u,\bar{u})\phi(0)\rangle$ satisfies a second-order equation, which can be turned into a hypergeometric equation of the form (174) with parameters

$$\tilde{a} = \frac{3}{5}, \qquad \tilde{b} = \frac{4}{5}, \qquad \tilde{c} = \frac{6}{5},$$
(201)

for the function $g(u,\bar{u})$ defined as

$$\langle\phi(\infty)\phi(1)\phi(u,\bar{u})\phi(0)\rangle = |u|^{-4h_\phi}|1-u|^{-4h_\phi}g(u,\bar{u}).$$
(202)

One obtains a solution of the form

$$F(x,\bar{x}) = |16x|^{4/5} \left| \frac{1-\sqrt{x}}{1+\sqrt{x}} \right|^{8/5} \left[ \widetilde{X}_1 \left| {}_2F_1(3/5,4/5;6/5|u) \right|^2 \right.$$
$$\left. + \widetilde{X}_2 \left| u^{-1/5} {}_2F_1(3/5,2/5;4/5|u) \right|^2 \right]. \tag{203}$$

## E Quantum Ising chain in an imaginary magnetic field

We consider the Hamiltonian:

$$H = -\frac{1}{2}\sum_{j=1}^{L} \left( \lambda\, \sigma_j^x \sigma_{j+1}^x + \sigma_j^z + ih\sigma_j^x \right), \tag{204}$$

with periodic boundary conditions, with $h$ and $\lambda$ real, in the regime $0 < \lambda < 1$.

Within the usual inner product all operators $\sigma_j^a$ are self-adjoint, and it is clear that $H$ is not (the matrix representing $H$ in the usual basis is symmetric but not real). Alternatively one can work with a different hermitian form, namely

$$\langle \Phi, \Psi \rangle = \langle \Phi | P | \Psi \rangle, \qquad P = \prod_{j=1}^{L} \sigma_j^z. \tag{205}$$

According to this hermitian form - which is not definite positive - the Hamiltonian density $\sigma_j^x \sigma_{j+1}^x + \sigma_j^z + ih\sigma_j^x$ (and therefore $H$ itself) is self-adjoint.

With the usual inner product, note that $PHP = H^\dagger$, so $P$ maps right eigenvectors to left eigenvectors. In particular in the $\mathcal{PT}$-unbroken phase, we have

$$H|r_0\rangle = E_0|r_0\rangle, \qquad H^\dagger|l_0\rangle = E_0|l_0\rangle, \qquad P|r_0\rangle \propto |l_0\rangle, \tag{206}$$

where $|r_0\rangle$ is the ground state. Then $|r_0\rangle = |l_0\rangle$ iff $|r_0\rangle$ is an eigenstate of $P$. For small systems ($L = 1, 2$) one can check analytically that this is not the case. We have observed numerically that this trend persists for larger systems. A curious observation is that for a single site ($L = 1$), the Hamiltonian is not diagonalizable at the transition : the two lowest eigenvalues $E_0$ and $E_1$ merge into a non-trivial Jordan block. It would be interesting to study whether this is also the case for larger systems, as it would suggest some logarithmic behavior in the continuum.

Despite being non-hermitian, the eigenvalues of $H$ are either real, or they appear in pairs of complex conjugates $(E, E^*)$. This can be seen using $\mathcal{PT}$ symmetry [51], or simply by noting that after the unitary similarity transformation $\widetilde{H} = UHU^\dagger$ with $U = \prod_{j=1}^{L}\exp(i\pi\sigma_j^z/4)$, one gets a real, nonsymmetric operator $\widetilde{H}$.

For $h = 0$, the Hamiltonian is Hermitian, and thus its spectrum is real. The regime $0 < \lambda < 1$ and $h = 0$ corresponds to the (anisotropic limit of) the 2d Ising model in the high-temperature phase, where the correlation length $\xi$ is finite. When $h$ is increased while $\lambda$ is kept constant, the ground state and first excited energies remains finite, up to a threshold value $h_c(\lambda, L)$, where they "merge" into a complex conjugate pair: see Fig. 6. The point $h_c$ corresponds to the vanishing of the partition function in the 2d Ising model. In the scaling limit, it converges to a critical point $h_c(\lambda, L) \to h_c(\lambda)$, called the Yang-Lee edge singularity.

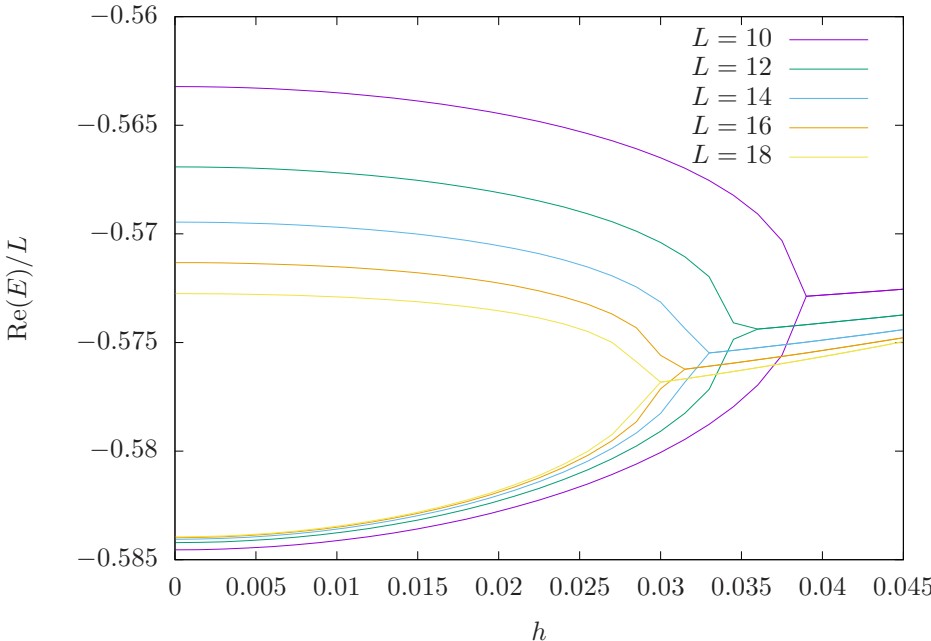

Figure 6: The two lowest energies in the Yang-Lee model (204) as a function of $h$, for $\lambda = 0.8$.

The finite-size study of the Yang-Lee edge singularity through the model (204) is rather subtle: for a given system size of $L$ sites, one should first determine the threshold value $h_c(\lambda, L)$, and then approach this value from below. We have computed numerically the one-interval ground state $N = 2$ Rényi entropy $S_2$ for the model (204) at $\lambda = 0.8$ and system sizes $L = 12, 14, 16, 18$ sites. The density matrix is defined as in the rest of the paper as $\rho = r_0 l_0^\dagger$, so the quantity we compute corresponds at criticality to (18). These numerical calculations lead to the following observations, depicted in Fig. 7. In the off-critical regime $h \ll h_c(\lambda, L)$, the entropy $S_2$ has a concave form. Then, when increasing the value of $h$ and approaching $h_c(\lambda, L)$ from below, the function undergoes a crossover to the convex form predicted by CFT (92). While not being positive, the entanglement entropy defined using $\rho = r_0 l_0^\dagger$ is surprisingly effective at detecting the phase transition.

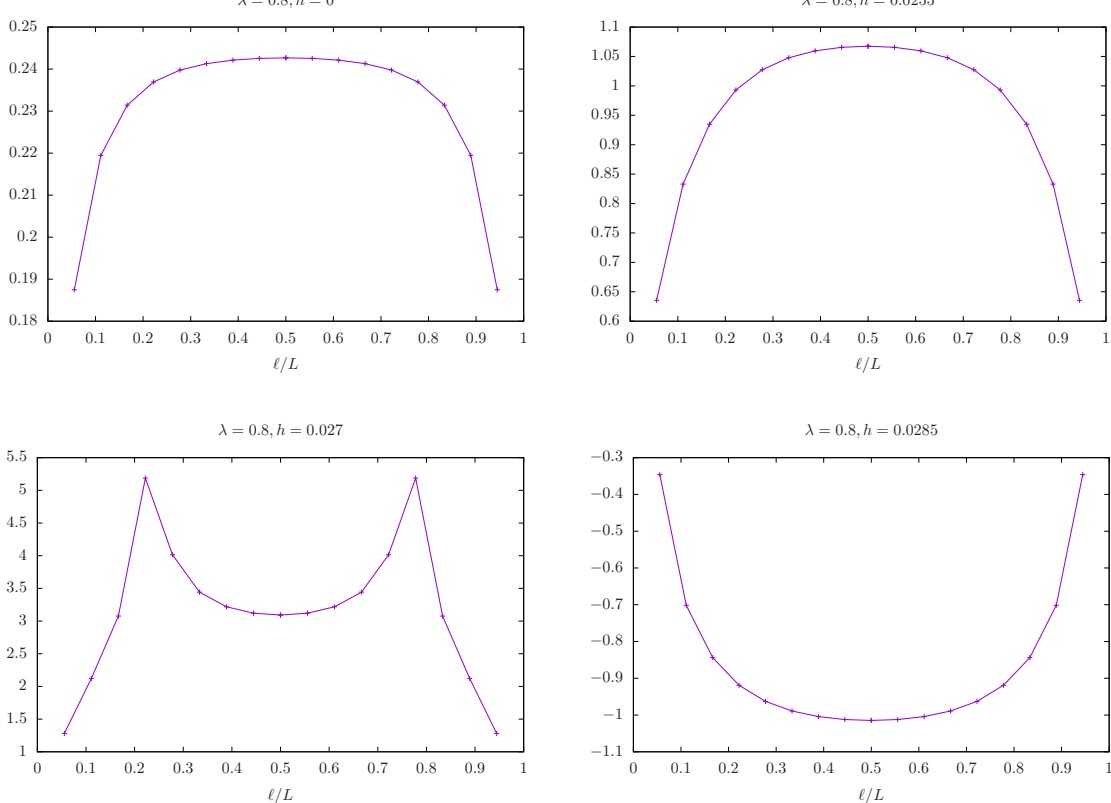

Figure 7: Crossover of the $N = 2$ ground state one-interval Rényi entropy in the Yang-Lee model (204) for $L = 18$ sites and $\lambda = 0.8$.

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
