# Peer review of "Entanglement entropies of minimal models from null-vectors"

_SciPost Physics, doi:SciPost Phys. 4, 031 (2018)_

## Round 4 · Referee Report · Anonymous (Referee 1) · 2017-11-13

Strengths

New analytic general method to compute Renyi entanglement entropies in arbitrary states, valid even for non-unitary models.

Weaknesses

Getting more and more involved as the Renyi index increases. Thus it is impossible to obtain closed forms for arbitrary N and take the replica limit for the von Neumann entropy.

Report

The authors introduce a new CFT formalism to calculate Rényi entropies of low order. This is based on the null-vector conditions on the twist fields in the cyclic orbifold, which induce a differential equation for their correlation functions that can be determined by bootstrap techniques. The authors apply this new methods to the simplest CFT, namely the non-unitary Yang-Lee model.
They recover results obtained with standard methods and generalise to CFT states in which the Rényi entropies were not known before.
Many of their results are checked against exact diagonalization of the RSOS model with continuum limit corresponding to the Yang-Lee CFT. The paper is extremely interesting and surely publishable, even if the authors must consider the following suggestions to improve the presentations.

Requested changes

  1. In Section 2, the most important concepts to perform the analytic calculation in the forthcoming sections are introduced. Understandably, none of the reported equations is derived. Yet, even for the experts in the field it would be very useful to have a reference close to the most complicated equations.

  2. In 2.7 the concept of "discrete Fourier transform" is introduced. It must be specified that this is a "finite Fourier transform" in replica space.

  3. The presentation of the figures 2 to 7 is rather badly done. There is not a line of text explaining them, but only some (not very clear) captions. It seems like one should only contemplate the figures and observe that data agree with the curve. For example, in one caption, one can read "Note that the variations of this entropy are incompatible with a two-point formula (5.23) with c replaced by the effective central charge ceff = 2/5." In a text the authors can discuss why at all one should think of comparing the data with ceff = 2/5 and not with their correct result. Furthermore, having so large figures one after the other without text does not help the readability. Maybe it would be better to have two figures with more panels (4+2 panels) or 3 figures with 2 panels each?

  4. I do not find the bibliography very accurate. Well known reviews on the entanglement are not mentioned (e.g. the one by Laflorencie in Phys. Rep., the Rev. Mod. Phys. by Eisert et al, etc.). The concept of twist field for the entanglement entropy was introduced in [5], but developed substantially by Cardy, Castro-Alvaredo and Doyon in J. Stat. Phys.130, 129 (2008). Many other possibly relevant references are missing. I leave the authors to dig in the literature and quote some of them. Furthermore, in the first page the citation to [13-17] is very misleading: These references deal with the calculation of the entanglement entropy for two disjoint intervals, while from the paper, one could erroneously understand that they refer to a single interval. At page 2, the authors write "When the subsystem A is the union of m disjoint intervals however, much less is known" where references to [13-17] would be really appropriate. In this respect, surprisingly the work Coser et al J. Stat. Mech. (2014) P01008 is not cited and it is the only one dealing with more than 2 interavals.

  5. There are a few typos in the manuscript. Here I list some that I spotted, but I recommend the authors to read very carefully the paper: a. On Eq. 2.5 a factor L is missing inside the logarithm. b. Page 19, cenvergence- > convergence c. Page 23, An lattice-> A lattice

  • validity: high
  • significance: good
  • originality: top
  • clarity: ok
  • formatting: acceptable
  • grammar: good

Author:  Thomas Dupic  on 2018-03-08  [id 226]

(in reply to Report 1 on 2017-11-13)

Thank you for your careful reading and comments,

  1. The corresponding references [26-28] can be found at the start of section 3 (former section 2).
  2. We have added "in replica space". However "discrete Fourier transform" is the correct terminology, rather than the ambiguous term "finite Fourier transform".
  3. We have improved the presentation of our numerical results accordingly.
  4. done.
  5. done.

---

## Round 4 · Referee Report · Olalla Castro-Alvaredo (Referee 2) · 2017-11-20

Strengths

1) This is a promising technique. It leads to differential equations for several non-trivial four-point functions involving twist operators (which play a prominent role in the context of entanglement) and other fields of the CFT under examination and could potentially lead to equations for higher point functions. 2) From the technical point of view, several non-trivial, new and interesting computations are presented which are valuable in the context of better understanding the properties of twist operators in CFT, their three-point couplings, correlation functions etc. 3) This new technique, consisting of finding differential equations for correlators of twist operators, could have applications in the study of entanglement measures in critical systems.

Weaknesses

1) The paper also has one major weakness. The authors have chosen to exemplify their general method for a non-unitary CFT. In doing so the authors have carried out various computations (analytical and numerical) which they claim represent various measures of entanglement, in particular Renyi entropies of an interval in a CFT at finite volume. 2) However the results that they obtain are clearly not Renyi entropies and therefore cannot be identified as such. A simple glance at their figures shows that what they are plotting cannot be Renyi entropies. 3) In particular the so-called "physical" Renyi entropy for N=2, shown in Figure 4, shows a function that is negative everywhere and which in addition decays as a function of the size of the interval, both properties which are in contradiction with any physically meaningful notion of entanglement of one interval in CFT. 4) Furthermore, this decay at short distance also contradicts the property that in the limit when the size of the interval is large and the size of the system is also large, the Renyi entropy should scale logarithmically with a positive coefficient (from their plots, it is clear this coefficient will be negative). This contradicts results for the entanglement of a semi-infinite interval in gapped systems, which should scale in the same way, up to a factor 2 in from of the logarithm (because of area law). 5) The authors claim that their results show that a previously proposed measure of entanglement for non-unitary systems is wrong. However, that proposal led to Rényi entropies which, as far as I can understand, have every desirable physically meaningful property. The same cannot be said of the current proposal.

Report

In this paper a new technique is proposed for the computation of the Renyi entanglement entropy in minimal models of CFT. The technique is based on employing the relationship between the entanglement entropy and correlation functions of twist operators and then applying the tools of CFT to obtain differential equations satisfied by these correlation functions (null-vector equations). This is a promising technique, whose main success in the present context, has been to allow the authors to obtain differential equations for several non-trivial four-point functions involving twist operators and other fields of the CFT under examination. I believe this is a good contribution to the field and, from the technical point of view, several non-trivial interesting computations are presented which are valuable in the context of better understanding the properties of twist operators in CFT, their three-point couplings, correlation functions etc. However, the paper also has one important weakness which is that the authors have chosen to exemplify their general method for a non-unitary CFTs and a non-unitary family of lattice models. In doing so the authors have carried out various computations (analytical and numerical) which they claim contradict existing results in the literature but which, from the physical point of view, are clearly wrong. To be more precise, what is wrong is the identification of certain correlatiors with the R\'enyi entropy (not the correlators themselves or their numerical evaluation). In the following I will provide a few basic reasons as to why this must be so. In addition, there is quite a bit of literature that is not cited and should be, so I will start with that.

  1. The authors cite a few papers in their introduction for their contribution to the theory of orbifolds. I think here or elsewhere the work V. Knizhnik, Analytic fields on Riemann surfaces. II, Commun. Math. Phys. 112(4) (1987) 56-590. should be cited as, in particular, it is the earliest work I know of where the conformal dimension of the twist field was computed.

  2. Nowhere in the paper (as far as I could see, maybe I missed it) do the authors ever acknowledge or mention where the connection between correlation functions of twist operators and the entanglement entropy was first pointed out. I think it would be fair and proper to mention this clearly somewhere, at least in the introduction (for instance, in the paragraph after their second equation in the introduction). As they probably know the first work where this type of connection was suggested was [5]. Here the fields were not called twist operators but it was observed that they could be associated with conical singularities in a replica theory. However, the derivations in this paper, particularly the one leading to the identification of the dimension of the twist field were not complete. They did not account for the fact that, in the replica theory, there are n copies of the stress-energy tensor, or, in other words, the central charge is nc (not c). As a consequence the authors of [5] found the wrong conformal dimension (missing a factor n). They compensated for this by claiming that the entanglement entropy of one interval is a two-point function of twist operators {\it raised to the power n. Today we know that this is not true. The correct derivation, by similar methods, was first presented in https://arxiv.org/abs/0706.3384. This paper also provided a more complete description of the twist operators, explicitly as local fields in a replica theory (conformal or not) and as symmetry fields (hence the name "branch point twist fields" given there) associated to an internal symmetry of the theory.

  3. Also, although this is a small detail, in the introduction, when the authors present their $c/3 \log \ell$ formula they cite [5,12] in this order. It would be more appropriate to cite [12] first as, after all, they were first. Also, before [5] there were several verifications of this formula for quantum spin chains, such as in the work [4]. So it would be more accurate to cite [12,4,5] for this formula (and possibly other works too).

  4. Also, the paper that showed that the entanglement entropy is a good measure of entanglement was the famous work: https://arxiv.org/abs/quant-ph/9511030 and should be cited.

  5. To my knowledge formula (2.29) appeared already in

Kac and Wakimoto, Acta Applicandae Math. 21 (1990) 3; https://arxiv.org/abs/q-alg/9610013 and in reference [24] and, as far as I know, in the context of entanglement it appeared first in https://arxiv.org/abs/1107.4280.

  1. In this paper, the authors compute several three-point couplings in CFT involving the twist operator (in particular, for the Lee-Yang model in appendix B). I believe many of these three-point couplings were already obtained in (see appendix A and D): https://arxiv.org/abs/1502.03275. This paper also contains a rather detailed description (for Lee-Yang) of the sort of fields that will feature in particular twist field OPEs, which I think is not entirely irrelevant in the present context. Also, a general prescription for the computation of three-point couplings involving branch point twist fields was given in appendix A.2 of https://arxiv.org/pdf/1006.0047.pdf and various properties of three-point couplings and particular ones (for the Ising model) were also worked out in https://arxiv.org/abs/1011.5482 and https://arxiv.org/abs/1112.1225.

The authors have a section on lattice models, where they discuss specifically the RSOS models. The entanglement entropy of a semi-infinite region in a certain class of RSOS models was first obtained in https://arxiv.org/abs/1509.04601.

7 Let me now come to my main criticism. I will illustrate it by examining the authors' results for the Lee-Yang model. Their main new claim is that the entanglement entropy of a single interval of length $\ell$ in the ground state (so the minimal energy state) of the Lee-Yang model at finite volume L is given by a four-point function involving a special type of twist field and the Lee-Yang field. The authors claim that this contradicts previous results that instead proposed the R\'enyi entropy to be a ratio of a two-point function of twist operators and $Z_1^N=\langle \phi(0) \phi(\ell)\rangle^N$ (seen as the norm of the ground state).

7.1 First of all, the claim above is not exact in the sense that the results of [25] are in infinite volume whereas the results of this paper are for finite volume. So one can not really compare both formulae directly to each other for they represent different things. However, it is obviously true that if one takes the formula of [25] and maps it to a finite volume situation (essentially by replacing $\ell$ by $\frac{L}{\pi}\sin\frac{\pi\ell}{L}$) then the two formulae disagree. This should be explained more clearly in the paper.

7.2 A priori, what the authors propose as a Rényi entropy seems reasonable. Let us consider a lattice model or spin chain model with a Lee-Yang critical point. This model is non-hermitian and as a consequence the ground state right and left eigenvectors of the Hamiltonian are not necessary equal. It is then natural to define a reduced density matrix given by $\rho=|\Psi_L\rangle \langle \Psi_R|$. The four-point function (2.40) that the authors propose as their replica partition function $Z_N=\mathrm{Tr}\rho^N$ is precisely the partition function that is obtained from this reduced density matrix in the continuum limit.

7.3 The problem with this is that, by construction, this reduced density matrix has some serious problems. The main obvious one is that it is not positive-definite, which in the context of entanglement is highly problematic. For one, this means that the Rényi and von Neumann entropies can be {\it negative}. Indeed, figure 4 shows precisely this, a set of values for the second Rényi entropy, all of which are negative. This problem could still perhaps be circumvented if some reason is found to add a constant to these results an shift the whole picture upwards (although, since the figure includes a comparison to numerics, I would expect there are no additional constants missing).

7.4 However, this is not even the worse problem this proposal has. A more serious problem is that the Rényi entropy {\it decays} with the interval length at small lengths. In fact all the functions the authors provide figures for have this feature. They look exactly like mirror reflections with respect to a horizontal line of the usual behaviour of the entanglement entropy.

7.5 Any reasonable physical interpretation of the entanglement entropy requires it to be a positive definite quantity, and if CFT is to describe its behaviour at large lengths then it should certainly increase in the region $\ell\ll L$, as it starts with the value 0 at zero interval length. So, at the very least, this is not a result that can just be presented without a serious discussion; and more likely, the results shown have little to do with Rényi entropies.

7.6 The question is, how do we resolve this issue? People working with non-hermitian quantum mechanical systems would probably suggest that the density matrix above is not the correct one. One should rather find a similarity transformation that maps the hamiltonian to a hermitian one and then re-define scalar products, operators and states in the resulting new Hilbert space where the density matrix should be positive definite. Indeed, I believe there are even some few papers on entanglement where people have done precisely this. The problem I see with this is that such similarity transformations, when they exist, are often highly non-local and often the resulting hermitian Hamiltonian is also non-local. So once the problem has been transformed in this way, the notion of a bi-partition on a Hilbert space where the local degrees of freedom are clearly defined is lost, and then it is hard to find a physical sense for any entanglement entropy that is based on this kind of density matrix.

7.7 As another possibility, and this is what was done in reference [25], one may decide that there are two fundamental requirements: a positive-definite density matrix and a clear identification of local degrees of freedom. This can be achieved by defining a R\'enyi entropy that is based on either $\rho_R=|\Psi_R\rangle \langle \Psi_R|$ or $\rho_L=|\Psi_L\rangle \langle \Psi_L|$, with ket and bra denoting, as usual, vector and dual in a properly defined Hilbert space. By doing this, one ensures that the density matrix is positive definite and one has scalar products in the original, standard Hilbert space. The result are true Rényi entropies that are positive definite, correctly measuring entanglement of a state in a Hilbert space. Recall that although the Lee-Yang model may have non-hermitian Hamitonian, there is, in the corresponding quantum chain, a proper Hilbert space, and the question of the scaling law of the entanglement of its ground state at the critical point is perfectly sensible. The proposal made in [25] is the continuum limit version of choosing the density matrix as above. This results in positive R\'enyi entropies that increase with interval length.

7.8 A simple question that the authors should certainly address is as to the infinite-volume limit, $\ell\ll L$. Clearly by scaling invariance the Rényi entropy will have the form $a \log \ell$ in this limit. What is the value of $a$ according to the authors? Is it positive? Looking at the figures presented here it looks negative to me.

7.9 Connected to the latter point, I believe the authors do not dispute the existing results for the entanglement entropy of a semi-infinite region. In the presence of a finite correlation length, it was shown in [25,26] and in the paper I mentioned in section 7 that the entanglement entropy scales as $\frac{c_{\mathrm{eff}}}{6}\log \xi$ (or as $\frac{c_{\rm{eff}}(n+1)}{12n}$ for the Renyi entropy), where $\xi$ is the correlation length. As is clearly the case for unitary CFT, one would expect also here that when $\ell$ and $L$ are large with $\ell\ll L$, the leading power law behaviour

$$ \frac{Z_n}{Z_1^n} \sim \ell^{2a} $$
where $a$ is the same power as in the case of the entanglement entropy of a semi-infinite system in a gapped model,
$$ \frac{Z_n}{Z_1^n} \sim \xi^{a}. $$
Does the function proposed by the authors satisfy this property? I haven't checked this precisely, but just looking at the shape of figure 4 (which is essentially reversed with respect to the figure in reference [25]) it would seem that for $\ell \ll L$ large we will never see a $\log \ell$ behaviour with a positive coefficient. And this means that the property above does not seem to be satisfied (while it clearly is for the proposal [25]).

  1. A couple of minor points now: In equation (2.7) I think $r$ has not been defined.
  2. In equation (2.11) the model indices are more restricted than the authors suggest. There is a relationship between the fractional part of $m$ and $r$ as the number of fields should be the same.
  3. What is the value of $r$ in equations (3.2) and (3.3). Before, the authors say there are only 2 values allowed but even so, this means there are several possible equations one could write depending on the choices of $r$. This should be better explained.
  4. In point 1 of section 5.3 I think the authors should write $|\Psi_L\rangle \langle \Psi_R|$ if they are talking about non-unitary models.
  5. In equation (5.18)-(5.20), shouldn't it be $\tau \tilde{\tau}$? And in (5.20) should they not divide by $Z_1^N$?

Requested changes

As a conclusion, although the authors have introduced a very promising technique, the results presented for the Yang-Lee model, which are technically correct, {\it are not Rényi entropies}. Thus appropriate qualifications should be given and comparison with the literature should be likewise improved. I would strongly recommend that, in the Yang-Lee context, the authors properly discuss the expected properties of the Rényi entropies and the problems one may encounter in non-unitary models, and that they qualify their calculations, clearly mentioning that none of the quantities calculated is actually a Rényi entropy. (And as a side remark, I do not think the authors should refer to the generic 4-point functions (2.41) as generalized Rényi entropies, as again these quantities generically have little to do with entropies or entanglement.)

  • validity: good
  • significance: good
  • originality: good
  • clarity: good
  • formatting: good
  • grammar: good

Author:  Thomas Dupic  on 2018-03-08  [id 225]

(in reply to Report 2 by Olalla Castro-Alvaredo on 2017-11-20)

Thank you for your careful reading, and the in depth comments,

  1. done.
  2. We have added the relevant references.
  3. done.
  4. done.
  5. done.
  6. done.
  7. We have added a substantial discussion addressing this question (section 2.3) and an appendix (E).
    7.1. Some results of arXiv:1405.2804 are also in finite volume, see Figure 2 therein. 7.2. We added a section (2.3) discussing this point. We argue that the choice of density matrix ($| \Psi_L \rangle \langle \Psi_R |$ or $| \Psi_R \rangle \langle \Psi_R |$) is to some extent a matter of definition.

We also note that most papers considering entanglement entropy for non-unitary models end up working with the same choice we made, namely $| \Psi_L \rangle \langle \Psi_R |$, whether they compute partition functions (as in arXiv:1509.04601), or they claim that $| \Psi_R \rangle = | \Psi_L \rangle$ (as in arXiv:1405.2804), or they simply make this choice (arXiv:1611.08506).

7.3 -- 7.6. It could be argued that non-positivity is not really an issue when dealing with non-unitary models. Underlying this is the choice of an inner product that this not positive-definite but such that the Hamiltonian density is self-adjoint. This is now discussed at length in section 2.3.

7.7. We have two issues with reference arXiv:1405.2804. First we disagree that left and right eigenvectors coincide (in finite size they clearly don't). Moreover if it were true, then our entropy would be positive, which is clearly not the case.

Second, the mapping of the R\'enyi entropy to a ratio of a two-point function of twist operators and $Z_1^N= \langle \phi(0) \phi(\ell) \rangle^N$ is incorrect. The issue is that $\langle \phi(0) \phi(\ell) \rangle$ is \emph{not} the norm of the ground state. In the CFT formalism, the state $| \phi \rangle$ has norm $1$ by definition. Only when doing a lattice calculation does one need to normalize states. The correct quantity is our equation (2.18). The collapse of the numerical data with different system sizes and the excellent agreement with our numerical calculations leaves very little doubt about this (see Figure 3). This is to be compared with the inadequate agreement in figure 2 of arXiv:1405.2804. First the authors did not demonstrate the collapse of numerical data using different system sizes, thus failing to establish that the data shown is indeed in the critical regime of the Yang-Lee model. Second, the agreement between the theoretical and numerical results is quite poor in the regime $\ell/L\sim 1/2$, i.e. the middle of the curve. However this is the regime in which the fit has to work best : for $\ell/L$ small the finite size effects are drastic (region $A$ being too small, only a few sites).

7.8. The $\ell \ll L$ behaviour of the Yang-Lee groundstate entropy for $N=2$ is all contained in the limit $x \to 1$ of function $G$ (5.12). The exponents associated to conformal blocks of the related function $F$ are given in (5.19). For generic integer $N \geq 1$, when $u \to v$ in (2.18), the dominant behaviour is determined by the OPE

$$ \tau_\phi(u,\bar u) \tau_\phi(v, \bar v) \sim |u-v|^{-4\widehat{h}\phi+2Nh\phi} \, \Phi(v,\bar v) \,, $$
which gives
$$ S_N \sim \left( \frac{N+1}{6N} c_{\rm eff} + \frac{2N}{N-1} h_\phi \right) \log|u-v| \,. $$
For instance, in the Yang-Lee model at $N=2$, one gets a negative prefactor $-11/5$, consistently with our numerical results.

7.9. Indeed in arXiv:1509.04601 the entropy of a semi-infinite interval within an infinite-volume system was studied for RSOS models in the massive regime. There it was found that the entanglement entropy diverges with the correlation length $\xi$ according to

$$S_N \sim \frac{c_{\rm eff}}{12} \frac{N+1}{N} \log \xi \,,$$
This calculation was done by mapping the R\'enyi entropy to the trace of some power of the corner transfer matrix, \emph{i.e.} a partition function on some multi-sheeted surface. In view of the above discussion, this calculation corresponds to our choice of density matrix\footnote{It was claimed in arXiv:1405.2804 that for RSOS models one always has $r_0 = l_0$ simply based on the reality of the transfer matrix. However this argument if flawed : while the transfer matrix is indeed real, it is not symmetric. Through a gauge transformation it can be made symmetric, but of course it will no longer be real. So there is actually no reason to expect that left and right eigenvectors coincide, and it is a simple matter of checking numerically to see that they do not agree in finite size.}, namely $\rho = r_0 l_0^{\dagger}$.

However the striking similarity between this result and the claim of arXiv:1405.2804 should not be taken as an argument in favor of the validity of the latter. It is true that for a unitary system the scaling of the entanglement entropy of a finite subsystem at criticality and that of an infinite subsystem with a small but finite mass gap are related. However the argument underlying this connection relies strongly on unitarity, and there is no reason to expect that this result can be extended to non-unitary systems, as was pointed out in arXiv:1509.04601.

  1. Corrected.
  2. Corrected.
  3. In the first equation, there are no Fourier modes, because it deals with the mother theory. In the second equation, the choice of $r$ is obvious from (3.18). Added a comment.
  4. Corrected.
  5. The normalisation is implemented by taking $\langle \psi_L|\psi_R \rangle=1$.

---

## Round 4 · Referee Report · Anonymous (Referee 3) · 2017-12-18

Strengths

1- old CFT method employed in a new context 2- both analytic and numerical analysis for a specific simple model

Weaknesses

1- difficult for the reader to extract the new results with respect to the previous literature 2- not clearly explained the problems in applying the method to other models

Report

The authors have employed the occurrence of null vectors in a rational conformal field theories (CFT)
to compute the entanglement entropies of an interval.
The analysis requires a highly non trivial understanding of the replica trick and its implementation in CFT.
The technical steps are exposed in a clear way.
Also a numerical analysis in the RSOS model is performed and agreement with theoretical formulas is obtained.

Requested changes

1- I suggest one main improvement. Since the benchmark of the method is the Yang-Lee model, which is simple but non unitary, I invite the author to introduce a dedicated discussion for the Ising model, with explicit claims about the new results found by the authors for this model.

2- Since the authors mentioned the case of disjoint intervals in various places, they could find worth mentioning also ref 1309.2189, where the twist field method has been employed for an arbitrary number of disjoint intervals in the Ising model and in the compactified boson.

  • validity: high
  • significance: good
  • originality: good
  • clarity: good
  • formatting: good
  • grammar: excellent

Author:  Thomas Dupic  on 2018-03-08  [id 224]

(in reply to Report 3 on 2017-12-18)

Thank you for your careful reading of the manuscript and your comments,

  1. For the Ising model essentially most results are already known, including multi-partite and excited states entropies. For this model we do not have new results. However we present an alternative derivation using our methods of the $N=2$ two-interval R\'enyi entropy (section 5.2), as well as the one-interval entropy for excited states (5.4 and 5.5).

  2. done

---

## Round 5 · Referee Report · Anonymous · 2018-3-8

Report

The authors properly took into account my suggestions, as well as the ones of the other referees. Although the list of changes and the answers to the referees should have been more detailed, I believe the paper is now ready for publication. I am sure that in spite of the technicality of the paper, it represents a genuine original and important piece of the literature on entanglement in CFT.

---

## Round 5 · Referee Report · Olalla Castro-Alvaredo · 2018-3-21

Strengths

The paper still has the same strengths I had pointed out in my original report. It proposes a method which seems to be new in the context of entanglement and which, as I said in my previous report, I think has merit and potential applications. The work is interesting beyond its application to entanglement as it more generally deals with the computation of multipoint functions of twist operators, some of which play an important role in the study of entanglement.

Weaknesses

Unfortunately, my two main criticisms from the original report still remain.

Not only have the authors not acted on my comment that it is intrinsically problematic to use the the name of "entanglement entropies" for functions that have negative values, negative curvature as functions of subsystem size, and are based on a reduced density matrix that is not positive-definite, but instead they have gone out of their way to insist that such functions are indeed entanglement entropies.

My other criticism that their so-called entanglement entropies did not seem to reduce to the expressions for two semi-infinite regions in the appropriate limit seems have been ignored.

Report

My previous report to the authors was very long and detailed and it entailed a considerable amount of work on my part. I am disappointed that the authors have not taken the time to properly and carefully respond to the report. Instead their description of changes made is so generic (and 7 lines long) that the only way for me to find out whether or not they followed my recommendations is to once again read the paper in great detail. I find this rather unprofessional and luckily in my experience, pretty rare.

Despite the above, I have had another good look at the paper and I appreciate that the authors have made an effort to adopt many of my suggestions, especially regarding references. They have also added additional sections to the paper to elaborate further on the special features of non-unitary theories and an appendix where they have done additional numerics on a non-hermitian quantum spin chain whose critical points describe the Lee-Yang edge singularity.

Despite these efforts the authors have not addressed my two main original criticisms. One of them was that it was intrinsically problematic to have an entanglement entropy function which is (in some cases) negative and which has also negative curvature as a function of the ratio of length scales. The authors have responded to this by confirming that indeed in some cases their reduced density matrices will not be positive-definite. Indeed, on page 7 of their paper they have now included the sentence

“The disadvantages of this construction are twofold. The main one is that the reduced density matrix (and hence the entanglement entropy) may not be positive. While this may seem like a pathological property, loss of positivity in a non-unitary system might be acceptable depending on the context and motivations”

It is good that the authors now write this explicitly but I happen to strongly disagree with their statement in the case of the entanglement entropies. Yes, non-unitary systems are different and some of their features seem counterintuitive but nonetheless admit a physical interpretation (for instance, complex energies may be interpreted as a sign of the presence of gain and loss in the system, which is a perfectly physical phenomenon). However, to my knowledge, there is no current physical interpretation of an“entropy” which is constructed from a non positive definite density matrix. Indeed, the whole interpretation of the entanglement entropy as a good measure of entanglement relies critically on the density matrix being positive definite, on it having positive eigenvalues whose sum is 1 and on it producing entanglement entropies whose minimum value is 0, corresponding to separable states. Therefore the authors’ statement that “negative values might be acceptable depending on context and motivation” is, in the case of the entanglement entropy, simply false.

Therefore, I think it is a shame the authors insist in calling their functions entropies. I have no problem with their statements that these functions have nice physical properties or even, as they say on their new appendix, that they may be used to precisely identify the phase transition in a particular physical system. That is all very good. These functions should be investigated. They are just not Rényi entropies.

The second point I had made in my previous report was that if they had any connection to any standard notion of entanglement entropy, then their functions should at least have a limit in which they reproduce the logarithmic behaviour of the entanglement entropy that has been found for two semi-infinite regions in gapped systems. Even if they study critical systems it is known since the work of Calabrese and Cardy that the scaling of the entanglement entropy of two semi-infinite gapped systems should mimic the scaling of the entanglement entropy of an interval of finite length within an infinite critical system. Given that the scaling of entanglement in gapped non-unitary systems has been identified in several publications and seems beyond doubt (it is not disputed by the authors either), it should be possible for the authors to take an infinite volume limit of their formulae and at least recover the correct known behaviour. As far as I can see in the current manuscript this point has not been addressed. I suspect that if it were addressed the results would be that their so-called entropies do not reproduce the correct infinite volume limit which would mean that non only they are not entanglement entropies, but they are not related to other quantities that are known to be.

In summary, although this paper has strengths and merit, the authors have unfortunately insisted on characterizing some of their results in an erroneous way. Although this may seem just a matter of words, I think it is an important matter. It is just the case that entanglement entropies are functions with a very particular set of properties and functions that do not share those properties, as interesting as they may be, cannot be called entanglement entropies. Based on these comments I am afraid my view is that the paper cannot be accepted for publication in SciPost in its current form.

  • validity: ok
  • significance: good
  • originality: high
  • clarity: high
  • formatting: excellent
  • grammar: good

Author:  Thomas Dupic  on 2018-03-22  [id 232]

(in reply to Report 2 by Olalla Castro-Alvaredo on 2018-03-21)

We really did appreciate the detailed comments you made on our paper, it was a great help in clarifying what we were trying to say and we did try to answer it thoroughly.

You may have missed the detailed answer to your first comment which was posted as a reply in the version 4 (https://scipost.org/submissions/1709.09270v4/) when the new version (v5) was added. It may not have been the right place to post it, as the new version now covers it by default, this is my first test of the Scipost submission system, I may have made a mistake and I apologize for it.

Sorry also for the short list of changes, it was just thought as a complement to the more detailed answers (and was not meant to be considered on its own).

We do take into account your new comments, and we will post an answer to them specifically as soon as possible

I repost here, for convenience, the first answer we made (which may answer some of your remarks) , sorry again for the misunderstanding.

Thank you for your careful reading, and in-depth comment,

  1. done.
  2. We have added the relevant references.
  3. done.
  4. done.
  5. done.
  6. done.

  7. We have added a substantial discussion addressing this question (section 2.3) and an appendix (E).
    7.1. Some results of arXiv:1405.2804 are also in finite volume, see Figure 2 therein. 7.2. We added a section (2.3) discussing this point. We argue that the choice of density matrix ($| \Psi_L \rangle \langle \Psi_R |$ or $| \Psi_R \rangle \langle \Psi_R |$) is to some extent a matter of definition. We also note that most papers considering entanglement entropy for non-unitary models end up working with the same choice we made, namely $| \Psi_L \rangle \langle \Psi_R |$, whether they compute partition functions (as in arXiv:1509.04601), or they claim that $| \Psi_R \rangle = | \Psi_L \rangle$ (as in arXiv:1405.2804), or they simply make this choice (arXiv:1611.08506).

7.3 -- 7.6. It could be argued that non-positivity is not really an issue when dealing with non-unitary models. Underlying this is the choice of an inner product that this not positive-definite but such that the Hamiltonian density is self-adjoint. This is now discussed at length in section 2.3.

7.7. We have two issues with reference arXiv:1405.2804. First we disagree that left and right eigenvectors coincide (in finite size they clearly don't). Moreover if it were true, then our entropy would be positive, which is clearly not the case.

Second, the mapping of the R\'enyi entropy to a ratio of a two-point function of twist operators and $Z_1^N= \langle \phi(0) \phi(\ell) \rangle^N$ is incorrect. The issue is that $\langle \phi(0) \phi(\ell) \rangle$ is \emph{not} the norm of the ground state. In the CFT formalism, the state $| \phi \rangle$ has norm $1$ by definition. Only when doing a lattice calculation does one need to normalize states. The correct quantity is our equation (2.18). The collapse of the numerical data with different system sizes and the excellent agreement with our numerical calculations leaves very little doubt about this (see Figure 3). This is to be compared with the inadequate agreement in figure 2 of arXiv:1405.2804. First the authors did not demonstrate the collapse of numerical data using different system sizes, thus failing to establish that the data shown is indeed in the critical regime of the Yang-Lee model. Second, the agreement between the theoretical and numerical results is quite poor in the regime $\ell/L\sim 1/2$, i.e. the middle of the curve. However this is the regime in which the fit has to work best : for $\ell/L$ small the finite size effects are drastic (region $A$ being too small, only a few sites).

7.8. The $\ell \ll L$ behaviour of the Yang-Lee groundstate entropy for $N=2$ is all contained in the limit $x \to 1$ of function $G$ (5.12). The exponents associated to conformal blocks of the related function $F$ are given in (5.19). For generic integer $N \geq 1$, when $u \to v$ in (2.18), the dominant behaviour is determined by the OPE

$$ \tau_\phi(u,\bar u) \tau_\phi(v, \bar v) \sim |u-v|^{-4\widehat{h}_\phi+2Nh_\phi} \, \Phi(v,\bar v) \,, $$

which gives

$$ S_N \sim \left( \frac{N+1}{6N} c_{\rm eff} + \frac{2N}{N-1} h_\phi \right) \log|u-v| \,. $$

For instance, in the Yang-Lee model at $N=2$, one gets a negative prefactor $-11/5$, consistently with our numerical results.

7.9. Indeed in arXiv:1509.04601 the entropy of a semi-infinite interval within an infinite-volume system was studied for RSOS models in the massive regime. There it was found that the entanglement entropy diverges with the correlation length $\xi$ according to

$$S_N \sim \frac{c_{\rm eff}}{12} \frac{N+1}{N} \log \xi \,,$$

This calculation was done by mapping the R\'enyi entropy to the trace of some power of the corner transfer matrix, \emph{i.e.} a partition function on some multi-sheeted surface. In view of the above discussion, this calculation corresponds to our choice of density matrix\footnote{It was claimed in arXiv:1405.2804 that for RSOS models one always has $r_0 = l_0$ simply based on the reality of the transfer matrix. However this argument if flawed : while the transfer matrix is indeed real, it is not symmetric. Through a gauge transformation it can be made symmetric, but of course it will no longer be real. So there is actually no reason to expect that left and right eigenvectors coincide, and it is a simple matter of checking numerically to see that they do not agree in finite size.}, namely $\rho = r_0 l_0^{\dagger}$.

However the striking similarity between this result and the claim of arXiv:1405.2804 should not be taken as an argument in favor of the validity of the latter. It is true that for a unitary system the scaling of the entanglement entropy of a finite subsystem at criticality and that of an infinite subsystem with a small but finite mass gap are related. However the argument underlying this connection relies strongly on unitarity, and there is no reason to expect that this result can be extended to non-unitary systems, as was pointed out in arXiv:1509.04601.

  1. Corrected.
  2. Corrected.
  3. In the first equation, there are no Fourier modes, because it deals with the mother theory. In the second equation, the choice of $r$ is obvious from (3.18). Added a comment.
  4. Corrected.
  5. The normalisation is implemented by taking $\langle \psi_L|\psi_R \rangle=1$.

---

## Round 5 · Referee Report · Anonymous · 2018-4-3

Strengths

1- old CFT method employed in a new context
2- both analytic and numerical analysis for a specific simple model

Weaknesses

1- difficult for the reader to extract the new results with respect to the previous literature
2- not clearly explained the problems in applying the method to other models

Report

The authors have employed the occurrence of null vectors in a rational conformal field theories (CFT)
to compute the entanglement entropies of an interval.
The analysis requires a highly non trivial understanding of the replica trick and its implementation in CFT.
The technical steps are exposed in a clear way.
Also a numerical analysis in the RSOS model is performed and agreement with theoretical formulas is obtained.

---

## Round 5 · Author Response

We thank the three referees for their careful reading of the manuscript and for their comments.

---

## Round 5 · List of Changes

- the first section was separated in two parts
- A subsection, specifically dedicated to non-unitary model was added in the first section to better explain our choices.
- Two examples were added, both apply the method of the paper to the entropy of a two-intervals subsystem . The first example (in subsection 4.1) concern the Yang-Lee model, while the second (in subsection 5.2 ) focus on the Ising model.
- The figures associated with the numerical results have been reordered.
- There is a new appendix (E) dedicated to the spin-chain representation of the Yang-Lee model .
- A part of the calculation originally explained in section 4. was displaced to the appendix (appendix B).

---

## Editorial Decision

published